# Astrocyte morphogenesis is dependent on BDNF signaling via astrocytic TrkB.T1

**Leanne M Holt[1,2], Raymundo D Hernandez[2,3], Natasha L Pacheco[1], Beatriz Torres Ceja[2], Muhannah Hossain[2], Michelle L Olsen[1,2]\***

[1]Department of Cell, Developmental, and Integrative Biology, School of Medicine, University of Alabama at Birmingham, Birmingham, United States; [2]School of Neuroscience, Virginia Polytechnic and State University, Blacksburg, United States; [3]Graduate Program in Translational Biology, Medicine, and Health, Virginia Polytechnic Institute and State University, Blacksburg, United States

**Abstract** Brain-derived neurotrophic factor (BDNF) is a critical growth factor involved in the maturation of the CNS, including neuronal morphology and synapse refinement. Herein, we demonstrate astrocytes express high levels of BDNF's receptor, TrkB (in the top 20 of protein-coding transcripts), with nearly exclusive expression of the truncated isoform, TrkB.T1, which peaks in expression during astrocyte morphological maturation. Using a novel culture paradigm, we show that astrocyte morphological complexity is increased in the presence of BDNF and is dependent upon BDNF/TrkB.T1 signaling. Deletion of TrkB.T1, globally and astrocyte-specifically, in mice revealed morphologically immature astrocytes with significantly reduced volume, as well as dysregulated expression of perisynaptic genes associated with mature astrocyte function. Indicating a role for functional astrocyte maturation via BDNF/TrkB.T1 signaling, TrkB.T1 KO astrocytes do not support normal excitatory synaptogenesis or function. These data suggest a significant role for BDNF/TrkB.T1 signaling in astrocyte morphological maturation, a critical process for CNS development.

**\*For correspondence:**
molsen1@vt.edu

**Competing interests:** The authors declare that no competing interests exist.

## Introduction

Astrocyte maturation is a crucial developmental process for normal CNS function. In the rodent cortex, astrocyte maturation takes place largely during the first 2–4 postnatal weeks. Importantly, this includes morphological maturation wherein immature astrocytes elaborate their processes and infiltrate the neuropil with fine, terminal, leaflet processes (*Bushong et al., 2004*). These leaflet terminals represent important functional structures, allowing cell–cell communication with neighboring astrocytes and enwrapping of synapses—where astrocytes participate in neurotransmitter uptake and synapse development and stabilization (*Farhy-Tselnicker and Allen, 2018*; *Oberheim et al., 2012*). Underscoring the morphological complexity of these cells, estimates indicate a single mature rodent astrocyte encompasses between 20,000–80,000 µM$^3$ of domain space (*Bushong et al., 2002*; *Halassa et al., 2007*), associates with 300–600 neuronal dendrites (*Halassa et al., 2007*), and contacts more than 100,000 individual synapses (*Freeman, 2010*). The maturation period of astrocyte morphogenesis coincides with neuronal synaptic refinement (*Freeman, 2010*; *Morel et al., 2014*) and differential expression of key genes associated with mature astrocyte functions, such as *Slc1a2* (Glt1), *Kcnj10* (Kir4.1), and *Aq4* (Aqp4) (*Clarke et al., 2018*; *Molofsky and Deneen, 2015*; *Molofsky et al., 2012*; *Morel et al., 2014*; *Nwaobi et al., 2014*). While the time course of astrocyte morphogenesis is well defined, few studies have attempted to identify molecular signals guiding astrocyte morphogenesis and maturation. To date, three mechanisms have been identified: Fibroblast Growth Factor (FGF)/Heartless signaling (*Stork et al., 2014*), glutamate/mGluR5 signaling (*Morel et al., 2014*), and contact-mediated neurexin/neuroligin (*Stogsdill et al., 2017*).

Brain-derived neurotrophic factor (BDNF) is a critical growth factor in the development, maturation, and maintenance of the CNS. Its role in neuronal cell growth, differentiation, morphology, and synaptogenesis via TrkB receptor signaling is well characterized (*Autry and Monteggia, 2012*; *Fenner, 2012*; *Park and Poo, 2013*). In the CNS, TrkB has two main isoforms. The full-length receptor, TrkB.FL, possesses a tyrosine kinase domain that autophosphorylates with BDNF binding, and a truncated receptor, TrkB.T1. While TrkB.T1 lacks the canonical tyrosine kinase domain, BDNF binding to this receptor is thought to signal through a RhoGTPase inhibitor and the phospholipase C (PLC) pathway (*Deinhardt and Chao, 2014*; *Fenner, 2012*). Dysregulation of BDNF/TrkB signaling has been implicated in multiple neurological and neurodevelopmental disorders (*Bolaños and Nestler, 2004*; *Merighi et al., 2008*; *Park and Poo, 2013*). However, a role for BDNF in the developmental maturation of astrocytes has not been investigated.

Here for the first time we demonstrate that *Ntrk2*, the gene that encodes BDNF's receptor is highly enriched in astrocytes during the critical period of astrocyte morphological maturation. RNA sequencing and qPCR reveal that in vivo astrocytes predominately express the truncated TrkB (TrkB. T1) receptor. TrkB.T1 receptor expression mediates increased astrocyte morphological complexity in response to BDNF in vitro, and TrkB.T1 KO astrocytes in vivo remain morphologically immature with significantly reduced cell volumes and morphological complexity. TrkB.T1 KO astrocytes exhibit dysregulation of genes associates with perisynaptic mature astrocyte function. Importantly, astrocyte-specific conditional knockout of TrkB.T1 also results in decreased morphological complexity. Finally, co-culture studies indicate TrkB.T1 KO astrocytes do not support normal synaptic development or function. Together, these data suggest a significant role for BDNF/TrkB.T1 signaling in astrocyte morphogenesis and indicate this signaling may contribute to astrocyte regulation of neuronal synapse development.

## Results

### Astrocytes express high levels of truncated TrkB.T1 mRNA

RNA sequencing was performed on cortical astrocytes that were acutely isolated from late juvenile (PND 28) mice via magnetic separation (n = 5 replicates) (*Holt and Olsen, 2016*). Postnatal day 28 was chosen based on reports indicating astrocytes are at the end of their morphological maturation period at this stage of development (*Bushong et al., 2004*; *Morel et al., 2014*). Analysis of this sequencing data revealed *Ntrk2,* the gene encoding BDNF's high affinity receptor TrkB, to be in the top 20 of all protein coding RNA's (#18) detected. The two TrkB isoforms are distinguishable given that the full-length receptor contains exons for the tyrosine kinase domain, while the truncated TrkB. T1 receptor lacks this domain but has an additional exon (exon 12) not found in the full length receptor (*Figure 1A*). Therefore, isoform-specific transcript expression was assessed and revealed that PND28 astrocytes predominately express the truncated isoform, with nearly 90% of all *Ntrk2* expression in cortical astrocytes attributed to TrkB.T1 (151.91 + /- 9.18 FPKM for NM_008745, 19.30 + /- 4.45 FPKM for NM_001025074) (*Figure 1B*).

This result prompted us to evaluate total, full length, and truncated *Ntrk2* mRNA expression in astrocytes relative to other CNS cell populations. Sequential isolation of oligodendrocytes, microglia, astrocytes, and neurons was performed as we have previously described (*Holt and Olsen, 2016*; *Holt et al., 2019*) again in PND 28 mice. Cellular purity was confirmed via qPCR by evaluating cell type specific gene expression (*Figure 1—figure supplement 1*). Oligodendrocytes were excluded for subsequent analysis due to lack of cellular purity (*Figure 1—figure supplement 1*). QPCR analysis of total and isoform-specific *Ntrk2* mRNA expression indicated total TrkB (primer detects both isoforms, gray in *Figure 1A*) was most highly expressed in astrocytes, relative to neurons or microglia. TrkB.FL is the predominant isoform expressed by neuronal populations (*Figure 1D*), while astrocytes predominantly expressed the truncated TrkB.T1 (*Figure 1E*). We next assessed TrkB isoform-specific expression across astrocyte development. QPCR analysis indicated low expression of the full length receptor regardless of time point. In contrast, expression of the truncated receptor was highest during the juvenile period (PND 28, *Figure 1F*) relative to young (PND 8) and adult (PND 60) astrocytes when total availability of BDNF peaks in the cortex (*Figure 1G*). Intriguingly, this time period correlates with the height of astrocyte morphological maturation.

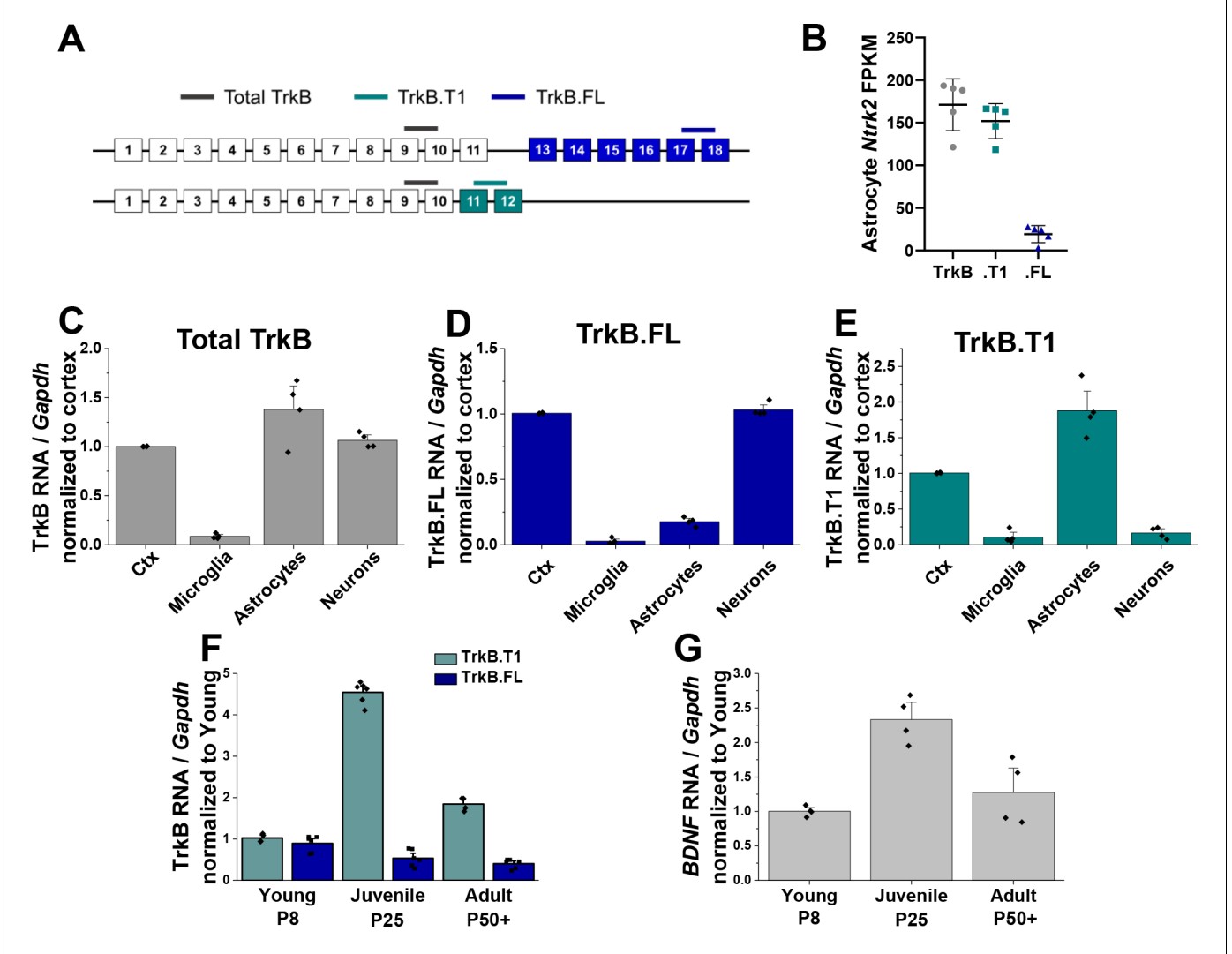

**Figure 1.** Astrocytes express high levels of truncated TrkB during astrocyte morphogenesis. (A) Cartoon representation of *Ntrk2* isoform exons and primers utilized. Total TrkB was probed with exons across both isoforms, with isoform-specific exons utilized to probe TrkB.FL (blue) and TrkB.T1 (teal) (B) Total and isoform-specific *Ntrk2* RNA expression in juvenile (PND 28) wildtype astrocytes. The majority of Ntrk2 expression in astrocytes is attributed to the truncated TrkB.T1 receptor isoform. Quantitative PCR analysis of acutely isolated CNS cell populations from juvenile (PND 28) mice for (C) overall *Ntrk2*, (D) TrkB.FL, or (E) TrkB.T1 mRNA expression, normalized to matched whole cortex. (F) *Ntrk2* receptor isoform mRNA expression in astrocytes across development (PND 7, PND 25, PND 50+). Expression of TrkB.T1 is highest in juvenile animals, when (G) Bdnf mRNA is highest in cortical tissues. Data represented as mean +/- SEM, n = 3–6 animals.

The online version of this article includes the following figure supplement(s) for figure 1:

**Figure supplement 1.** Validation of isolated CNS populations.

## Novel serum-free primary astrocyte cultures

To test a direct effect of BDNF/TrkB on astrocyte morphogenesis we turned to a novel in vitro astrocyte culture system. Here, astrocytes were acutely isolated from postnatal day 3–6 pups utilizing a previously published MACS sorting technique (*Holt and Olsen, 2016*; *Kahanovitch et al., 2018*), with the modification of elution and plating in a serum-free, defined media (*Figure 2A*). Cellular purity of the cultures was verified via qPCR, with mRNA levels of *Gfap* that are comparable to age-matched cortex but nearly undetectable levels of microglial, oligodendrocyte, OPC, and neuronal gene expression (*Figure 2B*) at both 7 and 14DIV. TrkB mRNA and protein expression was additionally verified, with similar levels of TrkB.T1 mRNA as age-matched in vivo astrocytes (*Figure 2C*). Importantly, in vitro astrocytes exhibited a developmental upregulation of TrkB.T1 mRNA at 14DIV,

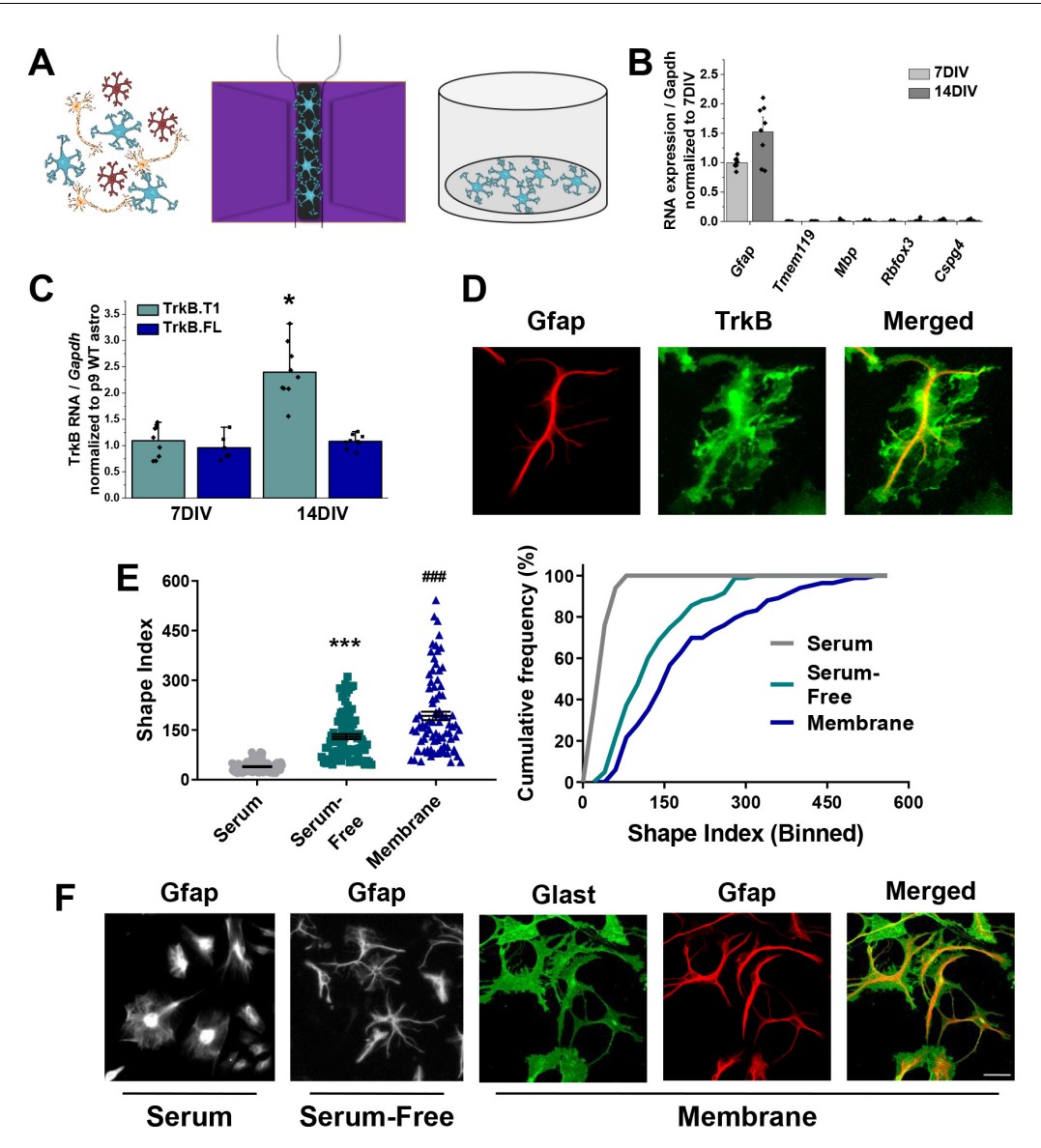

**Figure 2.** Novel serum-free primary astrocyte culture yields morphologically complex astrocytes. (A) Cartoon representation of magnetic separation of astrocytes for culture in serum-free, defined media. (B) Quantitative PCR data from cultured astrocytes at 7 and 14 DIV demonstrates purity of cultured cells. (C) mRNA expression of *Ntrk2* isoforms in cultured astrocytes compared to age-matched acutely isolated astrocytes demonstrates a developmental upregulation of TrkB.T1 expression. (D) Representative image of GFAP and TrkB immunofluorescence shows localization of TrkB to astrocytic membrane. (E) Shape index (SI) and cumulative frequency analysis of astrocytes cultured in serum-containing or serum-free media and membranous immunolabeling demonstrates increased cellular complexity in serum-free conditions. (F) Representative images of GFAP+ astrocytes cultured for 14DIV in the presence or absence of serum, and representatives images of membrane (Glast)/GFAP staining. Data represented as mean + /- SEM, n = 3–6 cultures, with two wells collected per culture. Each data point in E represents an individual cell.

similar to the upregulation seen juvenile astrocytes. Immunofluorescent co-staining of astrocytes with TrkB and GFAP revealed TrkB protein expression in astrocytes, with localization to astrocytic membrane (*Figure 2D*).

To examine astrocyte morphological complexity in vitro, we utilized the Shape Index (SI); this equation, given as perimeter$^2$/area – 4π, relates area and perimeter of a cell to a circle such that a perfect circle receives an SI of 0, and increasingly complex cells have correspondingly larger SI values. Notably, culturing astrocytes in serum free media resulted in 3.43-fold more complex astrocyte

as assessed by Shape Index of GFAP immunostained cells (t (198)=13.39, p<0.0001; *Figure 2E–F*). A cumulative frequency analysis was performed to account for the large range in the datasets, and demonstrates a significant right shift in serum-free astrocytes (D = 0.75; p<0.001). We additionally utilized a combination of Glast and Ezrin immunocytochemistry to demarcate both the membrane (Glast) and the fine, peripheral processes (Ezrin) in subsequent experiments. Glast—a membrane localized glutamate transporter—is highly expressed in astrocyte populations (*Kondo et al., 1995*) and Ezrin—a member of the ERM protein family—links the plasma membrane to the actin cytoskeleton, and has been previously demonstrated to be localized to peripheral astrocyte processes (*Derouiche and Frotscher, 2001*; *Lavialle et al., 2011*). The visualization and quantification of astrocyte membrane, which accounts for upwards of 85% of in vivo astrocytic volume, results in more physiologically relevant information. Unsurprisingly, comparison of Shape index quantification from membrane (Glast) staining revealed a 1.5 fold increased cellular complexity relative to intermediate filaments (GFAP) staining, (t(164) = 4.183, p<0.001; *Figure 2E–F*).

## BDNF induces an increase in astrocyte morphological complexity

Given the high levels of astrocytic TrkB expression during a period of astrocyte morphological maturation we next evaluated a role for BDNF on astrocyte morphology. Astrocytes were isolated and cultured as described above, and experiments performed after 14 DIV. Wildtype (WT) astrocytes were exposed to 10 ng, 30 ng, or 100 ng BDNF for 24 hr, followed by paraformaldehyde fixation. These concentrations were chosen based upon their previous use in investigating BDNF's effects on neurons (*Ji et al., 2005*; *Kline et al., 2010*) and astrocytes (*Ohira et al., 2007*). Experiments confirmed that Glast and Ezrin targets did not change expression following BDNF exposure (*Figure 3— figure supplement 1A*). Shape Index complexity analysis revealed BDNF-treated astrocytes showed a 2-fold increase in average astrocyte morphological complexity after exposure to 30 ng BDNF (F(3, 214)=7.047; p=0.001; *Figure 3A–C*). Tracings of WT and TrkB.T1 used for Image J analysis are shown in (*Figure 3—figure supplement 1B*). Cumulative frequency analysis additionally demonstrated a right shift at 30 ng BDNF (H(4) = 28.93; p=0.006), indicating a BDNF-induced increase in astrocyte morphological complexity. We confirmed this finding with similar experiments performed in cells stained with GFAP to visualize astrocyte branch processes. SI quantification revealed a significant 1.4-fold increase in average astrocyte morphological complexity after exposure to 30 ng BDNF (*Figure 3—figure supplement 1B–D*). Given that 30 ng BDNF exposure increased both astrocyte process and total cellular complexity, all following experiments were performed with this concentration.

## BDNF's effects are mediated via the truncated TrkB receptor

We next performed two loss of function experiments to ascertain the specificity of BDNF/TrkB.T1 effects on astrocytes. Current pharmacological TrkB receptor antagonists do not specifically target the truncated TrkB.T1 receptor. Therefore, we utilized TrkB-Fc to scavenge BDNF from the culture media. TrkB-Fc mimics the binding site of TrkB, allowing it to bind to BDNF and prevent BDNF binding to endogenously expressed TrkB receptors (*Guo et al., 2012*). WT 14DIV astrocytes were treated with 30 ng BDNF, and one hour later 2 ug TrkB-Fc additionally added. SI quantification of cellular complexity 24 hr later demonstrated that scavenging BDNF inhibited the increase in astrocyte morphological complexity (H = 70.41; p=0.98; *Figure 3D–F*).

To determine the necessity of TrkB.T1 receptor signaling in astrocytes we utilized a TrkB.T1 KO mouse model (*Dorsey et al., 2006*). We first validated specific loss of the truncated receptor in both tissue and astrocyte cultures. Both qPCR and western blot analysis of whole cortex and isolated astrocytes demonstrated loss of TrkB.T1, specifically (*Figure 3—figure supplement 2A–D*), with no change in TrkB.FL expression in isolated astrocytes (*Figure 3—figure supplement 2B*; F (3,16) = 0.08245; p = 0.8665). Importantly, sequential isolation of astrocytes, neurons, and microglia demonstrated TrkB.T1 KO mice do not exhibit a compensatory upregulation of the full length TrkB isoform (*Figure 3—figure supplement 2C*). Western blot quantification additionally confirmed this finding (*Figure 3—figure supplement 2D,E*). Astrocytes were isolated and cultured from male pups as described above. Quantitative PCR and immunocytochemistry experiments demonstrated that these cultures do not express the truncated TrkB receptor at the mRNA or protein level (*Figure 3— figure supplement 2F,G*). Note, the TrkB antibody used for these experiments recognizes both

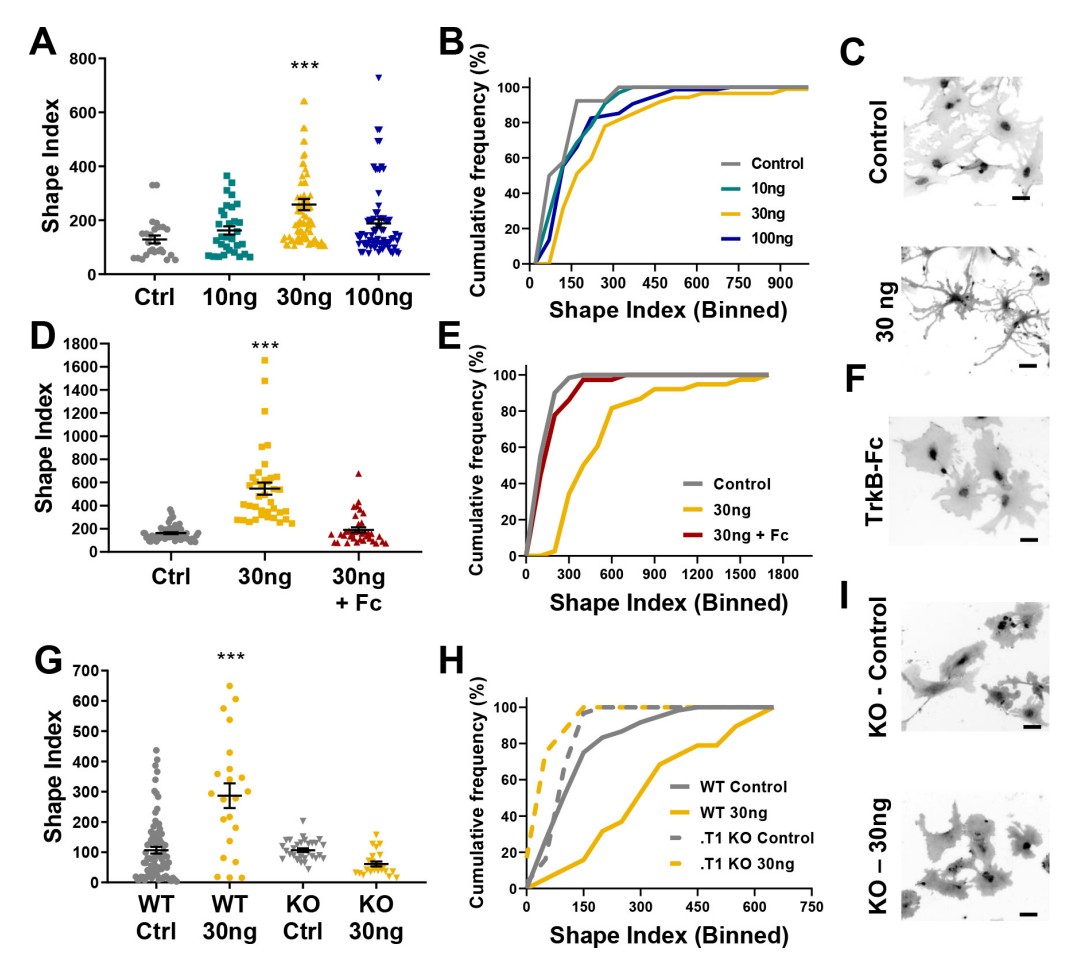

**Figure 3.** Astrocyte morphological complexity is sensitive to BDNF/TrkB.T1 signaling. (A) SI analysis and (B) cumulative frequencies following exposure to varying concentrations of BDNF for 24 hr. (C) Representative images of control and 30ng-treated astrocytes. (D, E) Complexity analysis of astrocyte morphology following scavenge of BDNF with TrkB-Fc, (F) representative image of TrkB-Fc scavenged astrocytes. (G–I) Morphological complexity analysis of cultured astrocytes lacking the TrkB.T1 receptor in the presence of BDNF. Data represented as mean + /- SEM, n = 3–6 cultures, with two wells collected per culture. Each symbol represents an individual cell. Scale bars indicate 20 microns. *p<0.05, **p<0.01, ***p<0.0001. Each data point represents an individual cell.

The online version of this article includes the following figure supplement(s) for figure 3:

**Figure supplement 1.** Wildtype astrocytes increase process branch complexity in response to BDNF.

**Figure supplement 2.** Validation of truncated TrkB isoform-specific deletion in TrkB.

isoforms of TrkB, and therefore also demonstrates no compensatory upregulation of TrkB.FL in the TrkB.T1 KO cultured astrocytes. At 14DIV, WT and. T1 KO astrocytes were exposed to 30 ng BDNF for 24 hr, and cellular complexity determined. As before, WT astrocyte SI indicated an increase in cellular complexity in response to BDNF ($F_{(3, 155)} = 21.66$; p = 0.0001). However, SI quantification and cumulative frequency analysis revealed no difference in control- and 30ng-treated TrkB.T1 KO astrocyte cellular complexities ($F_{(3, 155)} = 21.66$, p = 0.39; H (4, 136) =4 0.88, p = 0.99, respectively; *Figure 3G–I*). Our data, therefore, suggests that BDNF signaling through TrkB.T1 increases astrocyte morphological complexity.

## In vivo loss of TrkB.T1 decreases astrocyte morphogenesis

The experiments above established that BDNF signaling through the TrkB.T1 receptor induces an increase in astrocyte morphological complexity in a simplified model system. We set out to examine astrocyte morphogenesis as an indicator of astrocyte developmental maturation in TrkB.T1 KO mice.

Astrocyte morphology was examined in WT and TrkB.T1 knockout male animals at PND 14, PND 28, and PND 50. Here the early time point represents a period in astrocyte development when astrocytes are considered morphologically immature (*Bushong et al., 2004*; *Morel et al., 2014*). Intracerebroventricular (ICV) injections of AAV2/5 GfaABC1D driven lck-GFP in postnatal day 0/1 pups allows for sporadic labeling of astrocytes throughout the brain (*Figure 4A,B*). Confocal z-stack images of layer II/III motor cortex astrocytes were acquired. Imaris surface reconstruction allows for the determination of the full astrocyte morphology, and is indicative of the amount of neuropil infiltration of the astrocyte peripheral processes (*Morel et al., 2014*). Two-way ANOVA revealed a significant effect of genotype ($F(1, 101) = 9.413$, $p<0.01$) and time ($F(2, 101) = 42.16$, $p<0.0001$), as well a significant interaction ($F(2, 101) = 5.363$, $p<0.01$). In line with previous reports, WT astrocyte volume increased by 2-fold between PND 14 and PND 28 ($p<0.001$), and another 1.3-fold between PND 28 and PND 50 ($p = 0.0021$) with a total 2.8-fold increase in complexity between PND 14 and PND 50 astrocytes ($p = <0.001$), indicative of normal morphological maturation (*Figure 4C–E*) (*Bushong et al., 2004*; *Morel et al., 2014*; *Stogsdill et al., 2017*). Cumulative frequency analysis also demonstrated a significant right shift in astrocyte volume between PND 14, PND 28, and PND 50 astrocytes ($H(4, 89)=32.44$; $p<0.001$; *Figure 4E*). In contrast, this increase in morphogenesis was lost in TrkB.T1 KO animals, with no significant difference between PND 14 and 28 astrocyte volumes ($p=0.9996$; *Figure 4E*). It is not until PND 50 that astrocytes in TrkB.T1 KO animals demonstrated an increase in complexity and by only 1.8-fold between PND 14 and PND 50 ($p<0.01$), and 1.5-fold between PND 28 and PND 50 astrocytes ($p=0.03$; *Figure 4D,E*). No difference between WT and TrkB.T1 KO astrocyte morphology was detected at PND 14 ($p = 0.9987$; n = 16 cells from 5 KO animals; n = 18 cells from 5 WT animals). By PND 28, TrkB.T1 KO astrocytes demonstrated a 30% reduction in volume compared to WT littermates (*Figure 4C,E*; $p = 0.003$; n = 29 cells from 6 WT animals, n = 27 cells from 7 KO animals). The reduction in volume persisted into PND 50 animals, with a 20% reduction in volume compared to WT littermates (*Figure 4D,E*; $p = 0.03$; n = 21 cells from 5 WT animals, n = 17 cells from 5 KO animals). Thus, our data suggest that BDNF signaling onto TrkB.T1 in astrocytes is an important pathway for normal astrocyte morphogenesis.

We next assessed cortical thickness and astrocyte number in male TrkB.T1 KO and wildtype littermate PND 28 mice (*Figure 4—figure supplement 1*). Due to the decrease in astrocyte complexity, we hypothesized that there may be a compensatory increase in the number of astrocytes and/or a decrease in cortical thickness. We utilized serial sectioning, stereological microscopy, and optical fractionator software to determine the number of astrocytes and neurons in the motor/somatosensory cortex. Using Sox9 and NeuN to demarcate astrocytes and neurons respectively, we found no significant difference in the number of astrocytes or neurons in the TrkB.T1 KO mice compared to WT littermates ($p>0.05$; *Figure 4—figure supplement 1*). The total and layer-specific thickness of the cortex was also assessed, and again we found no significant differences in cortical thickness in the TrkB.T1 KO mice compared to WT littermates ($p>0.05$; *Figure 4—figure supplement 1*).

## In vivo loss of TrkB.T1 results in aberrant astrocytic gene expression

The period of astrocyte morphogenesis overlaps with differential gene expression in astrocytes in the developing cortex (*Clarke et al., 2018*; *Molofsky and Deneen, 2015*). We thus next examined genes located perisynaptically that are associated with mature astrocyte functions (*Clarke et al., 2018*; *Molofsky and Deneen, 2015*; *Nwaobi et al., 2014*; *Regan et al., 2007*). Astrocytes were acutely isolated in juvenile males (PND 25) from WT and TrkB.T1 KO littermates as described above. QPCR analysis revealed decreased mRNA expression in specific gene sets differentially regulated in astrocytes during maturation (*Clarke et al., 2018*): *Kcnj10* (Kir4.1; 51.6%, $t(4) = 2.943$; $p=0.04$) and *Aq4* (Aqp4; 21.7%, $t(4) = 2.807$; $p=0.04$), with trending decrease in *Slc1a2* (Glt1; 33.1%, $t(6) = 2.242$; $p=0.056$) relative to WT littermate controls (*Figure 4F*). Of note, we found no difference in *Gja1* expression (Cx43; $t(4) = 0.5491$, $p=0.612$) across the two genotypes, in line with previous studies indicating stable expression after p15 in rodents (*Aberg et al., 1999*; *Iacobas et al., 2012*; *Prime et al., 2000*). Thus, in vivo loss of TrkB.T1 results in dysregulated expression of genes associated with mature astrocyte function.

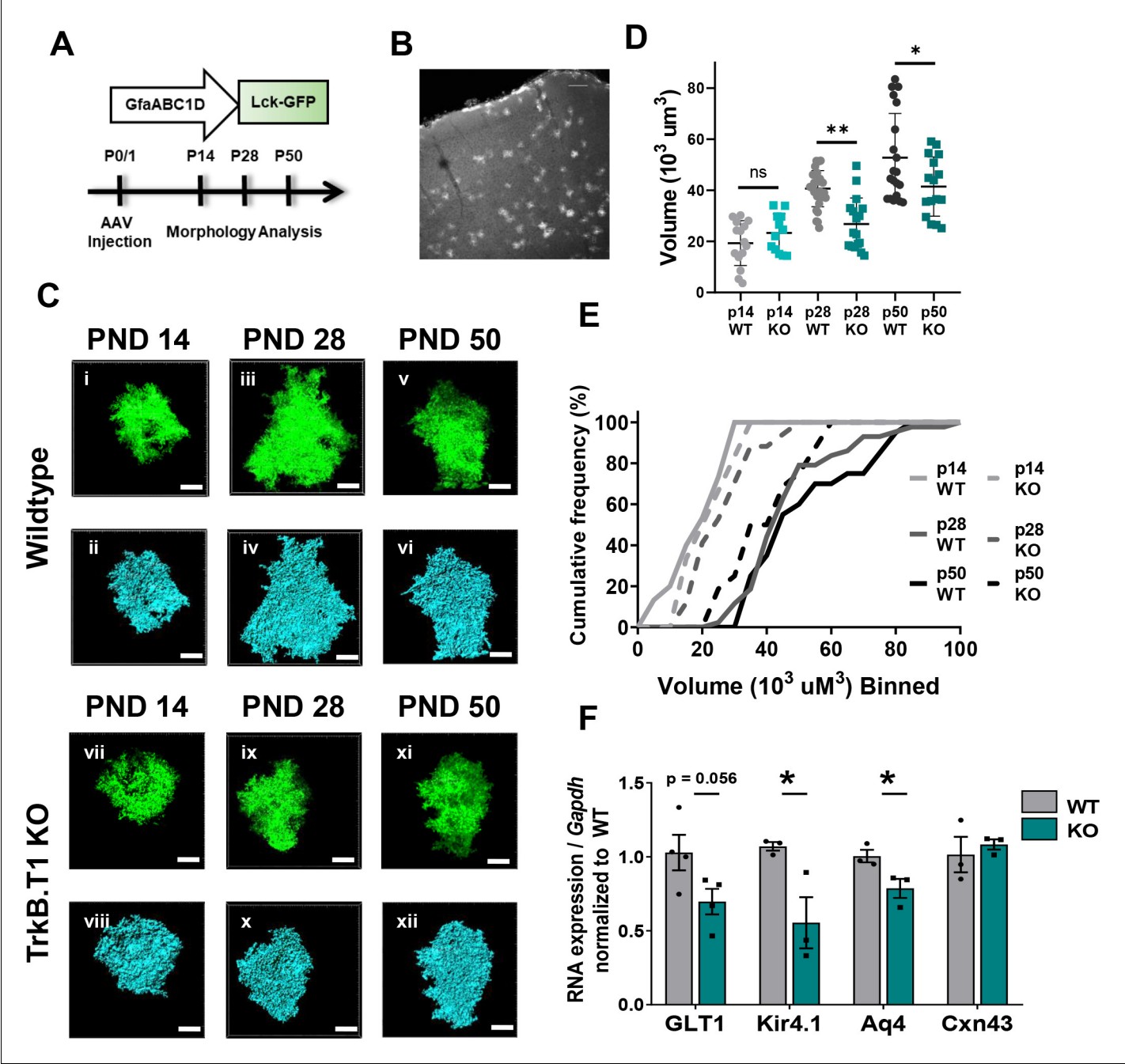

**Figure 4.** Loss of truncated TrkB.T1 in vivo leads to decreased astrocyte morphogenesis and an immature phenotype during maturation period. (**A**) Schematic of AAV injection and timeline of morphological analysis. (**B**) Representative image demonstrates sporadic, single astrocyte labeling in PND 28 animals following AAV injections. (**C**) Representative images of confocal (left, green) and Imaris surface reconstructions (right, blue) of astrocytes in (**i–iv**) wildtype and (**v–ix**) TrkB.T1 KO mice. Quantification of (**D**) astrocyte volume reconstruction and (**E**) cumulative frequency analysis demonstrates that prior to entering the morphological refinement phase (PND 14), wildtype and TrkB.T1 KO astrocytes exhibit no difference in their morphological complexity. However, by the end of the maturation phase (PND 28), TrkB.T1 KO astrocytes fail to increase their volume. It is only at PND 50 do TrkB.T1 KO astrocytes exhibit an increased morphological complexity. At PND 28 and PND 50 TrkB.T1 KO astrocytes exhibit a roughly 30% decrease in complexity compared to WT littermates. (**F**) QPCR analysis of WT and. T1 KO PND25 astrocytes demonstrates decreased mRNA expression of mature astrocytic genes Glt1, Kir4.1, and Aqp4, with no change in Cx43 expression. Data represented as mean + / - SEM, n = 5–7 animals for image analysis, with at least n = 3 cells per animal; n = 3 animals for qPCR analysis; *p<0.05, **p<0.01, ***p<0.0001. Each data point represents an individual cell in D, and an individual animal in F.

The online version of this article includes the following figure supplement(s) for figure 4:

**Figure supplement 1.** No change in astrocyte number in TrkB.T1 KO motor cortex.

## Astrocyte morphogenesis is dependent on astrocytic TrkB.T1 expression

We demonstrate thus far that TrkB.T1 KO astrocytes do not respond morphologically to exogenous BDNF and that loss of this receptor in vivo is associated with decreased astrocyte morphological maturation. An important limitation to the experiments performed above is 1) cultured astrocytes are removed from other cellular influence (i.e. endogenous BDNF expression) and 2) the utilization of a global TrkB.T1 mouse model whereby loss of the receptor is found in all cells. In order to query if—in otherwise normal environments—the loss of TrkB.T1 signaling specifically in astrocytes changes astrocytic morphology, we performed three experiments. First, astrocyte complexity was assessed in WT and .T1 KO astrocytes co-cultured with WT neurons. To this end, we developed a novel neuron-astrocyte co-culture model system. This system allows us to specifically and directly evaluate the interplay between astrocytes and neurons. Wildtype cortical neurons were cultured from p0/1 pups. After 3–5 days of recovery, astrocytes from p3-5 WT or TrkB.T1 KO animals were isolated via MACS and subsequently layered on top of the neurons. After 12-14DIV, cells were paraformaldehyde fixed and confocal images of astrocytes acquired. A combination of Glast and Gfap was used to visualize total (Glast) and branch complexity (Gfap) in the cultured astrocytes (*Figure 5A*). Sholl analysis was utilized to quantify the complexity of astrocytes co-cultured with WT neurons. Sholl analysis revealed that .T1 KO astrocytes were less complex overall compared to WT astrocytes as assessed by the Glast staining (*Figure 5b*; $F_{(1, 26)}$=17.88; p=0.03). Additionally, .T1 KO astrocytes exhibited decreased branch complexity (*Figure 5c*; $F_{(1, 18)}$=10.68; p=0.004) when co-cultured with WT neurons.

The above experiment suggests that TrkB.T1 expression in astrocytes is necessary for proper astrocyte complexity in a simplified system. We next examined astrocyte morphological complexity in a TrkB.T1 floxed mouse line (.T1$^{fl/fl}$) (*Dorsey et al., 2006*) utilizing AAV to specifically knockout the .T1 receptor in astrocytes alone. To this end, p0/1 mice pups were co-injected with two AAVs via ICV. AAV2/5 GfaABC1D.CRE-eGFP was used to drive Cre recombinase activity in TrkB.T1 floxed astrocytes, while AAV2/5 GfaABC1D.tdTomato serves as an internal control. Wildtype animals were also injected with the two AAVs as controls. Animals were collected at PND 28 (*Figure 5D*). We again examined astrocyte morphological complexity in layer II/III motor cortex via confocal z-stack imaging of individual astrocytes followed by Imaris surface reconstruction. Cre recombination in TrkB.T1 floxed astrocytes was identified with eGFP fluorescence. These astrocytes demonstrated a significant 20% reduction in volume compared to WT eGFP$^+$ astrocytes (t(21) = 2.407; p=0.0254; *Figure 5E–G*). Importantly, examination of the tdTom$^+$ control astrocytes revealed no significant difference in astrocyte complexity in the same WT and .T1$^{fl/fl}$ animals (t(23) = 0.7276; p=0.4742; *Figure 5—figure supplement 1*).

Finally, we generated an astrocyte-specific TrkB.T1 conditional knockout mouse model by crossing the TrkB.T1 floxed mice to the astrocyte-specific Aldh1l1Cre-ERT2 line. Breeding was maintained such that all offspring are homozygous TrkB.T1 floxed and either Cre$^+$ or Cre$^-$. All animals were intra-peritoneally injected with 20 µl (0.2 mg) 4-OH tamoxifen (TAM) once daily (*Zuo et al., 2018*) between PND 8–11 to specifically target astrocytes (*Figure 5H*). Sequential isolation of cortical microglia, neurons, and astrocytes was performed in PND 25–28 animals to validate loss of TrkB.T1 in astrocytes specifically. QPCR analysis revealed a significant reduction of TrkB.T1 expression in cKO astrocytes (85%; $F_{(1, 10)}$=12.80; p=0.0021; *Figure 5I*) in comparison to cWT astrocytes, with no difference in TrkB.T1 expression in neurons ($F_{(1, 10)}$=12.80; p=0.7372) or microglia. Importantly, we detected no significant change in TrkB.FL expression in astrocytes ($F_{(1,8)}$ = 0.1921; p=0.605), neurons ($F_{(1,8)}$ = 0.1921; p=0.761), or microglia (*Figure 5J*). ICV injection of the lck.egfp AAV was performed in p0/1 pups, followed by TAM injections between PND 8–11, and animals perfused at PND 28 (*Figure 5H*). Immunofluorescence using a TrkB.T1 specific antibody revealed decreased expression of TrkB.T1 expression in comparison to cWT littermates in the AAV-injected cKO astrocytes (*Figure 5K*). Confocal z-stacks of layer II/III motor cortex astrocytes were acquired (*Figure 5L*). Surface reconstruction of astrocyte volumes revealed that cKO astrocytes exhibited a significant 19% reduction in astrocyte complexity compared to cWT littermates (t(30) = 2.592; p=0.015; *Figure 5M, N*).

In a subset of cKO animals we also examined the same array of perisynaptically located astrocytic genes as above. To this end RNA was collected from the cortex of PFA fixed slices of PND25-28

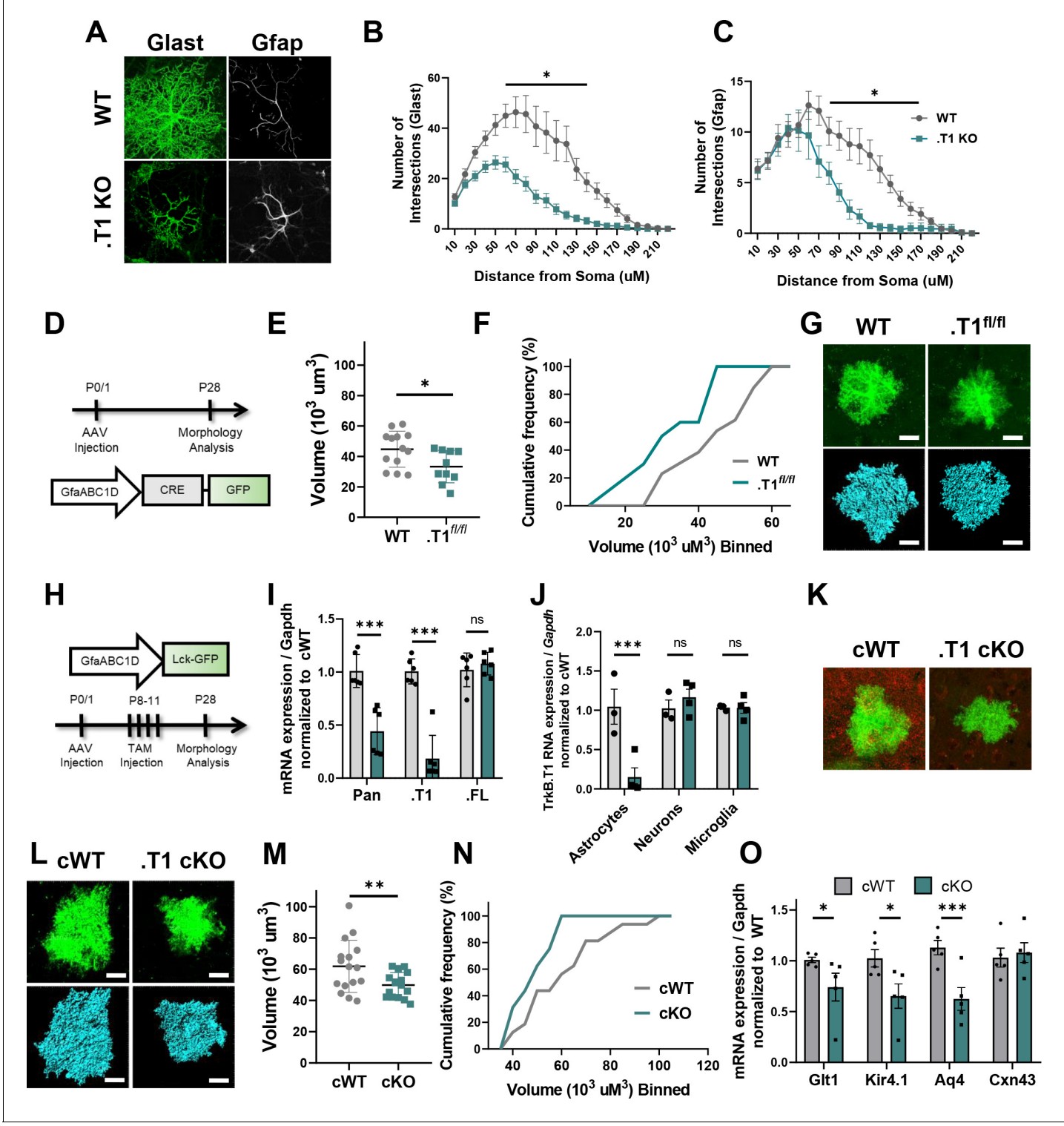

**Figure 5.** Specific loss of astrocytic TrkB.T1 results in decreased complexity. (A) Representative images of Glast and Gfap immunostaining of WT and .T1 KO astrocytes co-cultured with WT neurons. Sholl analysis revealed decreased complexity of .T1 KO astrocytes at the (B) total (Glast) and C) branch (Gfap) levels. (D) Schematic of timeline of AAV-Cre injections and morphology analysis. (E,F) Egfp-Cre+ astrocytes exhibited decreased astrocyte volume and complexity in. T1$^{fl/fl}$ animals compared to WT Egfp-Cre+ astrocytes. (G) Representative images of confocal (green, top) and Imaris reconstruction (blue, bottom). (H) Schematic of timeline to generate astrocyte-specific cKO animals. (I) QPCR analysis of isolated astrocytes revealed specific loss of TrkB.T1 isoform and no change in TrkB.FL isoform expression. (J) QPCR analysis of sequentially isolated astrocytes, neurons, and microglia reveal astrocyte-specific loss of TrkB.T1. (K) Representative confocal images of TrkB.T1 immunoreactivity in cWT and cKOs reveals loss of

*Figure 5 continued on next page*

*Figure 5 continued*

TrkB.T1 protein expression. (L) Representative images of confocal (green, top) and Imaris reconstructions (blue, bottom) of cWT and .T1 cKO astrocytes. (M, N) .T1 cKO astrocytes exhibited decreased morphological complexity in comparison to cWT astrocytes. (O) QPCR analysis of WT and .T1 cKO PND25 cortex demonstrates decreased mRNA expression of mature astrocytic genes *Slc1a2* (Glt1), *Kcnj10* (Kir4.1), and *Aq4* (Aqp4), with no change in *Gja1* (Cx43) expression. Data represented as mean + /- SEM, n = 4–7 animals for image analysis, with at least n = 3 cells per animal; n = 6 animals for qPCR analysis in I; three animals for qPCR analysis in J; *p<0.05, **p<0.01, ***p<0.0001. Each data point represents an individual cell in E and M, and an individual animal in I, J, and O.

The online version of this article includes the following figure supplement(s) for figure 5:

**Figure supplement 1.** No change in control, tdtomato+ astrocyte volume.

---

animals and qPCR performed. QPCR analysis revealed significantly reduced expression of the same genes found in the global TrkB.T1 KO mice. Specifically, cKO animals exhibited decreased expression of *Slc1a2* (Glt1; 25%; t(8) = 1.919; p=0.01), *Kcnj10* (Kir4.1; 35%; t(8) = 2.525; p=0.035), and *Aq4* (Aqp4; 44%; t(4) = 5.944; p=0.004) compared to cWT littermates. Similar to the global TrkB.T1 KO animals, no change was found in *Gja1* expression (Cx43; t(8) = 0.3699; p=0.7211) compared to cWT littermates. Thus, our data combined demonstrate that loss of the TrkB.T1 receptor in astrocytes, in an otherwise normal environment, results in reduced astrocyte morphogenesis and aberrant gene expression.

## Potential role for BDNF/TrkB.T1 astrocyte signaling on neuronal synapse development

The period of astrocyte morphological maturation overlaps with neuronal synaptogenesis, and a large body of work has characterized astrocytic contributions to this process. In order to query if BDNF/TrkB.T1 signaling onto astrocytes contributes to neuronal synapse development, we returned to our neuron/astrocyte co-culture system. Wildtype cortical neurons were again cultured from p0/1 mouse pups and allowed 3–5 days to recover, with subsequent plating of WT or .T1 KO astrocytes on top. After 8-12DIV, cells were paraformaldehyde fixed and confocal images of excitatory (VGlut1 +/PSD95+) synapses acquired (*Figure 6A*). Excitatory synapse number and function was evaluated due to extensive literature detailing astrocytic contributions to excitatory synaptogenesis (reviewed in *Allen and Eroglu, 2017*). Puncta Analysis quantification of colocalization of pre- and post-synapses (*Ippolito and Eroglu, 2010*; *Stogsdill et al., 2017*) revealed that, as expected, excitatory synapse numbers were increased in the presence of WT astrocytes (F(2, 121)=45.19, p<0.001; n = 39 cells None, n = 48 cells WT from N = 6 cultures) which did not occur in the presence of TrkB.T1 KO astrocytes (F(2, 121)=45.19, p=0.1542; n = 43 cells from N = 6 cultures; *Figure 6B–D*). Notably, further analysis revealed this effect appears to be influenced by differential consequences on pre- or post-synaptic sites. We observed a similar increase in the number of pre-synaptic puncta in excitatory synapses compared to WT astrocytes (F(2, 125)=18.88, p=0.1411; *Figure 6B*) with a significant reduction in post-synaptic partners (F(2, 125)=18.88, p<0.001; *Figure 6C*). This led to a significant reduction in co-localized and presumably functional synapses (*Ippolito and Eroglu, 2010*; *Stogsdill et al., 2017*). In a subset of experiments we repeated this analysis using a second set of known synaptic markers, Bassoon and Homer, and found similar effects of neuronal synaptic number (*Figure 6—figure supplement 1*). Neurons cultured with TrkB.T1 KO astrocytes did not exhibit the same increase in the number of co-localized synaptic puncta (F(2,38) = 2.8939; p=0.0015) in comparison to neurons cultured with WT astrocytes. In fact, the number of synapses did not differ from neurons cultured alone (F(2,38) = 2.8939; p=0.6069). Effects on synaptic function were also assessed. We utilized whole-cell patch-clamp to record miniature excitatory postsynaptic currents (mEPSC) in co-cultured neurons between 11–14 DIV (*Figure 6E,F*). In comparison to neurons co-cultured with WT astrocytes, neurons co-cultured with .T1 KO astrocytes demonstrated a reduced frequency (KS = 0.0456; p<0.001; n = 21 cells WT, n = 24 cells KO from N = 3 cultures; *Figure 6G*) but increased amplitude (KS = 0.116; p<0.0001; n = 21 cells WT, n = 24 cells KO from N = 3 cultures; *Figure 6H*). These studies suggest, at least in a simple model system, TrkB.T1 astrocytes do not support normal excitatory synaptic development and function.

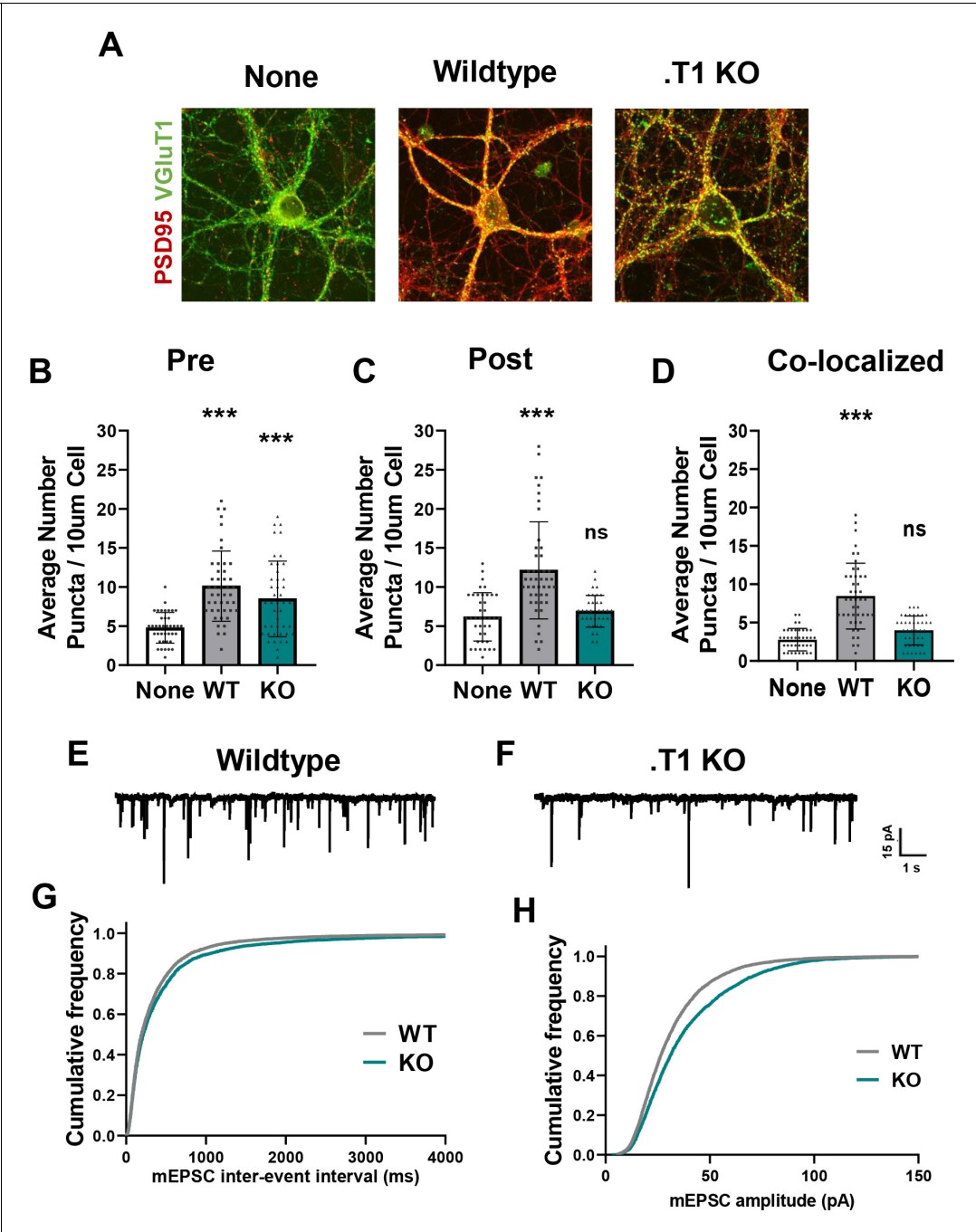

**Figure 6.** Astrocytes lacking TrkB.T1 receptor are unable to support neuronal synaptogenesis. (**A**) Representative images of neurons cultured in the presence of none, wildtype, or TrkB.T1 KO astrocytes. Excitatory synapses were visualized with VGlut1+/PSD95+ colocalization. Quantification of (**B**) pre-synaptic, (**C**) post-synaptic, and (**D**) colocalized, functional synapses of cultured neurons. Representative traces of mEPSCs in neurons co-cultured with (**E**) wildtype or (**F**). T1 KO astrocytes. Whole-cell patch recording demonstrated (**G**) reduced frequency but (**H**) increased amplitude of mEPSCs in neurons co-cultured with .T1 KO astrocytes compared to co-culture with WT astrocytes. Data represented as averages + /- SEM, n = 3–4 animals for qPCR; N = 6 cultures, with at least n = 4 cells per culture. *p<0.05, **p<0.01; *=compared to no astrocyte control. Each data point represents an individual cell in B-D.

The online version of this article includes the following figure supplement(s) for figure 6:

**Figure supplement 1.** Astrocytes lacking TrkB.T1 receptor are unable to support neuronal synaptogenesis.

## Discussion

Herein we have demonstrated for the first time that BDNF is involved in the maturation of a non-neuronal cell type. Within cortical astrocytes, TrkB.T1 receptor expression is in the top 20 of all protein-coding transcripts, with the highest expression during astrocyte morphological refinement. We developed and utilized a novel astrocyte culture paradigm to demonstrate that BDNF induces an increase in astrocyte morphological complexity, which is dependent upon the TrkB.T1 receptor. Importantly, in vivo astrocyte morphology is less complex with loss of the TrkB.T1 receptor in both global and astrocyte-specific murine models. In particular, the lack of TrkB.T1 prevented normal morphogenesis during the time of astrocyte morphological refinement, between PND 14 and PND 28 and in fact persisted into adulthood (PND 50), indicating BDNF/TrkB.T1 signaling contributes to astrocyte morphological maturation. Additionally, loss of the TrkB.T1 receptor in astrocytes resulted in aberrant expression of genes associated with mature astrocyte functions that are known to be developmentally regulated. Furthermore, BDNF signaling onto astrocytes appears to have consequences on neuronal synapse development, with decreased numbers and aberrant function of excitatory synapses in neurons co-cultured with TrkB.T1 KO astrocytes. These findings have broad implications, given the wealth of neurological disorders within which BDNF and, increasingly more often, astrocytes are implicated.

BDNF's role in CNS growth and maturation has been intensely studied, with a particular emphasis on the full length receptor and neuronal function. Here, we demonstrate that astrocytes express the highest levels of *Ntrk2* over other CNS cellular populations. Others have demonstrated that astrocytes express TrkB.T1 (*Ohira et al., 2007*; *Rose et al., 2003*). Prior to our work, however, TrkB receptor isoform expression in astrocytes across development and maturation was unknown. We therefore utilized a magnetic separation technique to isolate cortical astrocytes from young, juvenile, and adult animals. RNASequencing and qPCR analysis revealed that astrocytes predominately express the truncated TrkB.T1 isoform and intriguingly that TrkB.T1 demonstrated developmentally regulated expression. In particular, the highest expression of TrkB.T1 was found in juvenile (PND 28) animals. Indeed, RNASequencing in juvenile astrocytes revealed that TrkB expression is within the top 20 most highly expressed genes. In addition, we demonstrate that astrocytes express the highest levels of BDNF's receptor, TrkB, relative to neuronal and microglial populations. Publicly available RNA sequencing databases corroborate our data (*Zhang et al., 2014*), and, in fact, additionally demonstrate that human astrocytes express the highest levels of *Ntrk2* (*Kelley et al., 2018*; *Zhang et al., 2016*).

Truncated TrkB.T1 lacks the canonical tyrosine kinase signaling domain, and therefore has historically been presumed to act in a dominant negative capacity to prevent overactivation of BDNF signaling pathways (*Fenner, 2012*; *Klein et al., 1989*; *Middlemas et al., 1994*). We here have shown that BDNF induces an increase in astrocyte morphological complexity, and this effect is lost in TrkB.T1 KO astrocytes. Therefore, our data suggest an intrinsic and direct mechanism of action. Supporting this, in vivo dysfunction of astrocytic TrkB.T1 has been implicated in mediating neuropathic pain and motor dysfunction following spinal cord injuries (*Matyas et al., 2017*). BDNF application to cultured astrocytes resulted in a PLCy-IP3R mediated rise in calcium with kinetics different from cultured neurons (*Rose et al., 2003*). Additionally, the TrkB.T1 receptor has been found to co-immunoprecipitate with a RhoGTPase inhibitor, RhoGDIa in primary astrocyte cultures, and inhibition of RhoA increased the area occupied by cultured astrocytes (*Ohira et al., 2005*). Given RhoGTPases's known roles in regulating astrocytic cytoskeletal dynamics (*Zeug et al., 2018*), this presents a potential mechanism by which BDNF increases astrocyte morphological complexity in astrocytes. A potential interaction and/or regulation of RhoGTPases by TrkB.T1 may also offer insight into the BDNF dose-specific response we observed in cultured astrocytes. Time and dose dependent effects have been observed for many BDNF/TrkB.FL downstream signaling mediators, including phosphorylation of CREB, ERK, and Akt (*Cunha et al., 2009*; *Jia et al., 2007*; *Mamounas et al., 2000*), leading to alterations in neuronal cell survival, axonal sprouting, and enhancement or deficits to learning and memory behaviors (for review see *Kowiański et al., 2018*). To date, this has not been examined specifically for TrkB.T1. If TrkB.T1 mediates astrocyte morphological complexity via inhibition of RhoGTPases, then too little (10 ng/ml) or too much (100 ng/ml) inhibition could result in no discernible change in astrocyte morphology. Future work aimed at determining the primary downstream signaling pathways of TrkB.T1 are needed to elucidate the dose dependent changes in astrocyte

morphology we observed. It is unclear how BDNF is processed by astrocytes following binding to TrkB.T1. However, our data suggest TrkB.T1 plays an active, direct role in astrocyte biology.

Little is known regarding the mechanisms governing astrocyte morphogenesis and maturation. Thus far, three mechanisms have been identified. In *Drosophila*, FGF signaling through the Heartless receptor determines astrocyte domain size and infiltration into the neuropil (*Stork et al., 2014*). Pharmacological and genetic manipulations of mGluR5 in astrocytes reduced the developmental increase in astrocytic volume between PND 14 and 21 (*Morel et al., 2014*). Similarly, loss of neuroligins in astrocytes and/or their neuronal neurexin partner reduced astrocyte morphogenesis in the visual cortex by PND 21. Based on our findings, we propose BDNF signaling through truncated TrkB.T1 receptor as another mediator of astrocyte morphological maturation. Similar to others (*Morel et al., 2014*; *Stogsdill et al., 2017*), we found that wildtype astrocytes exhibited a 1.75-fold increase in volume during astrocyte morphological maturation. However, TrkB.T1 KO astrocytes did not exhibit normal morphogenesis. No difference in astrocyte volume between PND 14 and PND 28 in TrkB.T1 KO animals was found, and we only observed an increase in .T1 KO astrocyte complexity at PND 50. However, there is still a roughly 40% decrease in increased complexity compared to WT astrocytes between PND 14 and 50, indicative of a failure to properly undergo morphogenesis and maturation. In fact, at PND 28 there is a 30% decrease in astrocyte volume compared to mature wildtype littermates, which persists in PND 50 animals.

We did not observe a difference in astrocyte number by stereological microscopy across the motor cortex, nor did we observe a difference in cortical motor thickness. $Sox9^+$ astrocytes represented a mere 12% of total $DAPI^+$ cells in this region. Our observed difference in volume at PND 28 was ~30%, given the small number of astrocytes relative to total number of cells this change may not result in measurable differences in overall cortical volume. While not influencing overall cortical volume, the difference in astrocyte complexity may impact astrocyte infiltration into the neuropil, enwrapment of synapses and neuronal function. Future work using EM, specifically in the astrocyte specific knock out are needed to address this question. It should be noted that the global TrkB.T1 knock out animals demonstrate anxiety like behaviors (*Carim-Todd et al., 2009*). Studies in astrocyte specific TrkB.T1 KO are needed to determine if loss of TrkB.T1 in astrocytes contribute to this aberrant behavior.

Of note, in cultured astrocytes, control-TrkB.T1 KO astrocyte complexity did not differ in comparison to control-WT astrocytes, and we observed a reduced complexity in the. T1 KO astrocytes only following exposure to BDNF. While in vitro work allows for the specific manipulation of the cell culture media, it also removes cells from the influence of other CNS cell populations. The loss of signaling molecules from other cells, in this context the lack of BDNF present in the media, may explain our data. This assertion is also supported by our TrkB-Fc scavenge experiments whereby removal of BDNF from the culture media prevented the increase in astrocyte morphological complexity. Additionally, in co-culture experiments, TrkB.T1 KO astrocytes co-cultured with WT neurons exhibited a decreased complexity compared to WT astrocytes co-cultured with WT neurons. Of note, it is well characterized that neurons exert many effects on astrocytes. In fact, all of the three thus identified mechanisms governing astrocyte morphological maturation involve neuron-astrocyte communication.

An important limitation is the utilization of a global TrkB.T1 knockout mouse model to assess in vivo astrocyte morphology. We therefore devised three experiments to test astrocyte-specific TrkB.T1 contribution to astrocyte morphological complexity. First, we demonstrated that TrkB.T1 KO astrocytes exhibit decreased complexity compared to WT astrocytes in the presence of WT neurons. Second, we used an astrocyte-specific AAV to drive cre recombinase activity specifically in astrocytes in TrkB.T1 floxed mice. Importantly, we also expressed a control AAV driving only the tdTomato fluorescent reporter in these animals. We found that only the $Egfp^+$-Cre astrocytes from TrkB.T1$^{fl/fl}$ animals exhibited a change in astrocyte complexity, specifically a 20% decrease, in comparison to WT $Egfp^+$-Cre astrocytes. Within the same animals, there was no difference in complexity of the control, $tdTom^+$ astrocytes. Finally, we generated an astrocyte specific conditional knockout mouse model. Utilization of the Aldh1l1-creERT2 mouse line allows for temporal and spatial control of TrkB.T1 expression in astrocytes. We injected animals with TAM between PND 8–11 in order to examine astrocyte morphogenesis. We indeed saw a significant decrease in astrocyte complexity in the cKOs compared to cWT littermates at PND 28. Together, our data indicate that BDNF/TrkB.T1 plays an

important role in normal astrocyte morphogenesis and may act as a mediator of astrocyte morphological maturation.

Peripheral astrocyte processes (PAPs) are dramatically increased in complexity during the morphological refinement phase, and facilitate many known astrocyte biological functions. We therefore examined expression of known PAP-targeted genes associated with mature astrocyte function. TrkB.T1 KO astrocytes demonstrated decreased mRNA expression of Kir4.1, Glt1, and Aq4 in juvenile (PND 28) mice. Noteworthy, these genes are known to be developmentally regulated (*Furuta et al., 1997*; *Hsu et al., 2011*; *Nwaobi et al., 2014*; *Regan et al., 2007*; *Wen et al., 1999*). Another astrocytic PAP gene, *Gja1* (Cx43), did not change in expression. Interestingly, although Cx43 is associated with mature astrocyte function, previous work demonstrates mRNA expression stabilizes after PND 15 in rodents (*Aberg et al., 1999*; *Iacobas et al., 2012*; *Prime et al., 2000*). Importantly, in our astrocyte-specific TrkB.T1 conditional KO mice we found the same gene expression changes. This suggests that BDNF/TrkB.T1 signaling may regulate a specific set of genes in astrocytes, however, future work is needed to elucidate the exact mechanism.

Astrocytes actively contribute to synaptogenesis through release of astrocyte derived factors such as hevin (*Kucukdereli et al., 2011*), thrombospondins (*Christopherson et al., 2005*), SPARC (*Kucukdereli et al., 2011*), and glypicans 4/6 (*Allen et al., 2012*). Astrocyte enwrapment of synapses is additionally known to be regulated by neuronal activity, and can stabilize synapses following LTP (*Bernardinelli et al., 2014*). As TrkB.T1 KO astrocytes demonstrate decreased morphological complexity, we additionally investigated how BDNF signaling onto astrocytes may impact neuronal synapse number and function. To this end, we developed a novel astrocyte-neuron co-culture paradigm. Utilization of MAC sorting technique allows for separation and subsequent combination of different cellular subtypes, genotypes, and ages within cultures. We posit that this technique will be useful to many for investigations of cell-to-cell communication. We found neurons cultured in the presence of .T1 KO astrocytes exhibited decreased numbers of excitatory post-synaptic elements and an overall reduction in numbers of co-localized pre and post-synaptic puncta. We additionally found aberrant synaptic function of neurons co-cultured with .T1 KO astrocytes, with a decrease in frequency but increase in the amplitude of mEPSCs. This is particularly interesting as previous work demonstrated no change in basal synaptic transmission or LTP in the hippocampus, but found decreased neuronal arborization in the amygdala and anxiety-like behavior in 8 week old global TrkB.T1 KO mice (*Carim-Todd et al., 2009*). We utilized cortical neurons and astrocytes for all experiments, and future investigations are needed to determine regional specificity of BDNF/TrkB.T1 effects on astrocytes. The decreased frequency of mEPSCs observed is consistent with the decreased synapse number found in neurons co-cultured with .T1 KO astrocytes. We additionally observed an increase in mEPSC amplitude, which could result from increased AMPAR currents or increased pre-synaptic vesicular loading. Additionally, increased mEPSC amplitude is associated with synaptic scaling in response to reduced activity, and we demonstrate a concomitant reduction in the number of synapses and frequency of mEPSC. Similar patterns of synaptic dysfunction were found in hippocampal slice mEPSCs of neurodevelopmental disorder Rett Syndrome, with right shifted cumulative probabilities in both frequency and amplitude (*Calfa et al., 2015*). Intriguingly, aberrant BDNF expression and dysfunctional astrocytes are well characterized in Rett. Future work is needed to determine the mechanism of increased amplitude.

We demonstrated that scavenging BDNF from the media within an hour of exposure prevented the increase in cellular complexity 24 hr later, suggesting that BDNF must be actively present to elicit an increase in astrocyte morphological complexity. This experiment is particularly interesting given that following synaptogenesis, BDNF secretion from neurons is largely targeted to synaptic zones and is secreted in an activity-dependent manner (*Park and Poo, 2013*). While outside of the scope of this paper, the influence of BDNF on astrocyte morphological complexity may indeed extend into activity-dependent maintenance of astrocyte morphology and enwrapment of synapses. One study suggests that this may indeed occur, as siRNA knockdown of TrkB.T1 in adult rats leads to decreased ability of cortical astrocytes to modulate their morphology in response to neuronal activity (*Ohira et al., 2007*). These results also highlight that BDNF may be necessary for the maintenance of astrocyte morphology in adulthood.

Here we demonstrate BDNF's receptor, TrkB.T1, is highly enriched in cortical astrocytes, particularly during the period of astrocyte morphological maturation, and that BDNF/TrkB.T1 signaling in astrocytes plays a critical role in astrocyte morphogenesis and may play a role in proper astrocyte

maturation. Furthermore, proper neuronal synaptogenesis was lost with deletion of the TrkB.T1 receptor in astrocytes. Our studies suggest that BDNF/TrkB.T1 signaling is a novel unexplored pathway in the role of astrocytes in synapse development. Given the role of aberrant synapse development in neurological dysfunction, our results herein suggest astrocyte BDNF/TrkB.T1 signaling may contribute to neurodevelopmental disorders in which BDNF signaling is implicated.

# Materials and methods

## Key resources table

| Reagent type (species) or resource | Designation | Source or reference | Identifiers | Additional information |
|---|---|---|---|---|
| Strain, strain background (*M. musculus*, male) | Wildtype; WT | This paper | | See Materials and methods, Animals section |
| Strain, strain background (*M. musculus*, male) | TrkB.T1 KO;. T1 KO | Dr. Lino Tessarallo | | Global TrkB.T1 knockout mice |
| Strain, strain background (*M. musculus*, male) | Aldh1l1-cre/ERT2 | Jackson Laboratories | Stock #029655 | Astrocyte-specific TAM-inducible CRE mice line |
| Strain, strain background (*M. musculus*, male) | TrkB.T1$^{fl/fl}$;. T1$^{fl/fl}$ | Dr. Lino Tessarallo | | Floxed TrkB.T1 mice line |
| Transfected construct (*M. musculus*) | GfaABC1D.Lck.egfp | Addgene | Cat#105598-AAV5 | Membrane-tethered eGFP to visualize astrocyte morphology |
| Transfected construct (*M. musculus*) | GfaABC1D.Egfp-Cre | UNC Gene Therapy Center Vector Core | | Astrocyte-specific deletion of TrkB.T1 and visualization of astrocyte morphology |
| Transfected construct (*M. musculus*) | GfaABC1D.tdTomato | Addgene | Cat#44332-AAV5 | Control for viral-mediated deletion of TrkB.T1 in astrocytes; visualization of astrocyte morphology |
| Antibody | Rb-GFAP (polyclonal) | DAKO | Cat#Z0334 | 1:1000/Overnight |
| Antibody | Rb-Glast (polyclonal) | Abcam | Cat#ab416 | 1:500/Overnight |
| Antibody | Ms-Ezrin (monoclonal) | Sigma | Cat#E8897 | 1:500/Overnight |
| Antibody | Gp-VGLUT1 (polyclonal) | Millipore | Cat# AB5905 | 1:1000/Overnight |
| Antibody | Ms-PSD95 (monoclonal) | NeuroMab | Cat#75028 | 1:1500/Overnight |
| Antibody | Rb-TrkB (pan; polyclonal) | Millipore | Cat#07–225 | 1:1000/1 hr, WB; 1:1000/Overnight, IF |
| Antibody | Rb-Sox9 (monoclonal) | Abcam | Cat#ab185966 | 1:500/Overnight |
| Antibody | Ms-NeuN (monoclonal) | Millipore | Cat# MAB377 | 1:500/Overnight |
| Antibody | Rb-TrkB.t1 (polyclonal) | SantaCruz | Cat#sc119 | 1:750/Overnight |
| Sequence-based reagent | *Gfap* | Thermo Fisher Scientific | Cat#Mm 01253033_m1 | |

*Continued on next page*

*Continued*

| Reagent type (species) or resource | Designation | Source or reference | Identifiers | Additional information |
|---|---|---|---|---|
| Sequence-based reagent | Tmem119 | Thermo Fisher Scientific | Cat#Mm 01248771_m1 | |
| Sequence-based reagent | Mbp (myelin basic protein) | Thermo Fisher Scientific | Cat# Mm01266402_m1 | |
| Sequence-based reagent | Rbfox3 (NeuN) | Thermo Fisher Scientific | Cat# Mm01248771_m1 | |
| Sequence-based reagent | Cspg4 (NG2) | Thermo Fisher Scientific | Cat# Mm0057256_m1 | |
| Sequence-based reagent | Kcnj10 (Kir4.1) | Thermo Fisher Scientific | Cat# Mm00445028_m1 | |
| Sequence-based reagent | Slc1a2 (Glt1) | Thermo Fisher Scientific | Cat# Mm00441457_m1 | |
| Sequence-based reagent | Aq4 (Aquaporin-4) | Thermo Fisher Scientific | Cat# Mm00802131_m1 | |
| Sequence-based reagent | Cx43 (Connexin-43) | Thermo Fisher Scientific | Cat# Mm01179639_s1 | |
| Sequence-based reagent | Ntrk2 (TrkB, total) | Thermo Fisher Scientific | Cat# Mm00435422_m1 | Recognizes all isoforms; targets transmembrane spanning exons |
| Sequence-based reagent | Ntrk2 (TrkB.Fl) | Thermo Fisher Scientific | Cat# Mm01341761_m1 | Recognizes full length isoform; targets tyrosine kinase domain spanning exons |
| Sequence-based reagent | Ntrk2 (TrkB.T1) | Thermo Fisher Scientific | TCAAGTTGGCGA GACATTCCA | Recognizes truncated TrkB.T1 isoform; targets. T1 specific exon |
| Sequence-based reagent | Gapdh | Thermo Fisher Scientific | Cat#4352339E | |
| Peptide, recombinant protein | TrkB-Fc | R and D Systems | Cat#688-TK-100 | 2 ug |
| Peptide, recombinant protein | BDNF | Promega | Cat# G1491 | |
| Chemical compound, drug | (z)−4-hydroxytamoxifen; TAM | Tocris | Cat#3412 | 20 mg/ml stock concentration; 20 uL 1X day between PND 8–11 |

## Animals

All experiments were performed according to NIH guidelines and with approval from the Animal Care and Use Committee of the University of Alabama at Birmingham and Virginia Polytechnic Institute and State University. All animals were maintained on a reverse 12 hr light/dark cycle (lights on at 9pm, lights off at 9am) with food and water available *ad libitum*. Every effort was made to minimize pain and discomfort. Wild-type and $TrkB.T1^{-/-}$ and wild-type littermate (*Dorsey et al., 2006*) C57/B6 mice were used for these experiments. $TrkB.T1^{-/-}$ and TrkB.T1 floxed mice were a generous gift from Dr. Lino Tessarollo.

Astrocyte-specific conditional knockout mice were generated by crossing Aldh1l1-cre/ERT2 (Jackson Labs #029655) to TrkB.T1 floxed mice. Experimental animals were generated by crossing homozygous. $T1^{fl/fl}$ $Cre^+$ males to homozygous. $T1^{fl/fl}$ females such that all offspring were homozygous floxed, and either $Cre^+$ or $Cre^-$. Each experimental animal was injected to with 4-OH tamoxifen prior to knowing genotype. Tamoxifen was made to stock concentration of 20 mg/ml in 1:19 ethanol:

canola oil via sonication for 60 min at 50℃. Injections were administered intraperitoneally at 20 µL between PND 8–11.

## Cortical dissection and dissociation

Briefly, mice (young, postnatal day 7 + /- 1 days (PND 7), late juvenile mice (PND 28 + /- 3 days (PND 28) or adult mice (PND 60+ /- 10 days (PND 60) were anesthetized via $CO_2$ and decapitated. Whole cortex was microdissected in ice-cold ACSF (120 mM NaCl, 3.0 mM KCl, 2 mM MgCl, 0.2 mM CaCl, 26.2 mM NaHCO3, 11.1 mM glucose, 5.0 mM HEPES, 3 mM AP5, 3 mM CNQX) bubbled with 95% oxygen. Tissue was minced into 1 $mm^3$ pieces and dissociated for 15–30 min using Worthington Papain Dissociation Kit. Tissue was subsequently triturated until a single-cell suspension was achieved and filtered through a 70 µM filter.

## Astrocyte isolations

Astrocytes were acutely isolated as previously described (*Holt and Olsen, 2016*; *Kahanovitch et al., 2018*; *Stoica et al., 2017*). Following dissociation, microglia and myelin were first removed from the cell suspension. Cells were incubated for 15 min at 4℃ with 15 µL of Miltenyi Biotec's Myelin Removal Kit and Cd11b+ MicroBeads. The suspension was then applied to a prepped LS column, washed three times, and the flow-through collected. This flow through was subsequently used to isolate astrocytes utilizing Miltenyi Biotec's ACSA-2+ MicroBead kit. The cell suspension (in 150 µL 0.5% fatty-acid free BSA in PBS) was incubated at 4℃ for 15 min with 15–20 µL FcR blocker, followed by a 15 min incubation with 15–20 µL ACSA-2 microbeads. Cells were applied to a prepped LS column. Astrocytes were eluted from the LS column after three washes, with 5 mL buffer and the supplied plunger.

## Sequential CNS population isolations

Cells were acutely isolated as previously described (*Holt and Olsen, 2016*). Following dissociation, oligodendrocytes were isolated first with a 10 min incubation with 15 µL Myelin+ microbeads. Cells were applied to a prepped LS column, and washed 3x. All flow through was collected and utilized to isolate the subsequent cellular populations. Oligodendrocytes were eluted from the LS column after three washes, with 5 mL buffer and the supplied plunger. Microglia were isolated next, with a 10 min incubation with 15 µL Cd11b+ microbeads. Cells were applied to a prepped LS column, and washed 3x. As before, all flow through was collected and utilized to isolate the next cellular population. Microglia were eluted from the LS column after three washes, with 5 mL buffer and the supplied plunger. Astrocytes were subsequently isolated as described above. Finally, neuronal populations were isolated using Neuronal isolation kit. The flow through was again collected and used to isolate neurons utilizing Miltenyi Biotec's Neuron Isolation Kit. The cell suspension was incubated with 20 µL biotinylated antibodies for 10 min at 4℃, followed by a 15 min incubation with 20 µL anti-biotin microbeads. Cells were applied to a prepped LD column. Neurons were collected in the flow through of two washes.

## RNA isolation and qPCR

Total RNA was isolated using Ambion's PureLink RNA Mini Isolation kit according to the manufacturer's instructions. RNA samples designated for RNA Sequencing were eluted in 30 µL filtered, autoclaved Mill-Q water. Subsequently, 2 ng of RNA was reverse transcribed into cDNA using Bio-Rad's iScript kit or BioRad's iScript SuperMix. All cDNA was normalized to 350 or 500 ng (for BDNF mRNA assays) following conversion. The relative mRNA expression levels were determined using real-time quantitative PCR by General Taqman PCR master mix and TaqMan specific probes. Relative mRNA expression levels were determined by the ddCt method, with each normalization indicated where appropriate.

## RNA sequencing

RNA samples were tested for quality on the Agilent Tapestation 2200 (Agilent Technologies, Santa Clara, CA). The NEB Next rRNA Depletion Kit (NEB #E6310X) was used to process 250 ng of total RNA. RNA-Seq libraries (400 bp) were created using the NEBNext Ultra II Directional RNA Library Prep Kit for Illumina (NEB #E7760L). Samples were individually indexed using the NEBNext Multiplex

Oligos for Illumina (NEB #E6609S). Adapter ligated DNA was amplified in 13 cycles of PCR enrichment. Libraries were quantified with the Quanti-iT dsDNA HS Kit (Invitrogen) and qPCR. Library validation was performed on the Agilent 2200 Tapestation. Independently indexed stranded cDNA libraries were pooled and sequenced for 150 cycles with the Illumina NovaSeq 6000 S2 Kit. All samples were sequenced at 85–90 million read depth, paired-end 2 × 150 bp, and in reverse-stranded orientation.

## Bioinformatics analyses

Initial analyses (raw reads processing through read alignment) were run in the University of Alabama at Birmingham's Cheaha High Performance Computing (HPC) cluster environment. Raw RNA-Seq reads were concatenated (per R1 and R2 fastq read, respectively) and quality trimmed using Trim Galore! Version 0.4.3. Sequence quality of trimmed reads was inspected using FastQC (version 0.11.15). The STAR aligner (version 2.5.2) (*Dobin et al., 2013*) was used in the basic two-pass mode to align the trimmed reads to the iGenomes UCSC mm10 mouse genome. BAM files were sorted by coordinate, and indexed using SAMtools (*Li et al., 2009*) (version 1.3.1). To examine general gene expression levels, a gene counts table was created using featureCounts (*Liao et al., 2014*) (release 1.5.2) and used as input for DESeq2 (*Love et al., 2014*) (version 1.16.1) in the RStudio environment (version 3.4.1). Genes with a row sum less than 10 were excluded prior to differential gene expression analysis. Normalized counts were extracted for each biological replicate to calculate the average normalized counts per respective gene. For transcript expression analysis, the STAR-aligned BAM files were processed in the University of Alabama at Birmingham Galaxy platform (*Afgan et al., 2016*) using Stringtie (*Pertea et al., 2016*) (Galaxy tool version 1.3.3.1) as described in the recommended workflow, with minor modifications: 1) the reverse strand option was selected and 2) the iGenomes UCSC mm10 genome was used as the reference guide assembly data set for the first Stringtie run. *Ntrk2* transcript expression levels (FPKM) were extracted from the second Stringtie run's Assembled Transcripts output files per respective biological replicate. All detailed scripts used for these analyses are available upon request.

## Protein extraction and immunoblotting

Proteins were extracted by homogenizing samples in lysis buffer (1% sodium dodecyl solfate (SDS), 100 mM Tris(hydroxymethyl)aminomethane (Tris) buffer, pH 7.5), supplemented with protease and phosphatase inhibitors (Sigma), followed by two rounds of sonication for seven seconds. Lysates were subsequently centrifuged for 5 min at 16,000 x*g*. ThermoScientific's Peirce BCA assay was utilized to determine protein concentrations. Proteins were heated to 60°C for 15 min with 2x loading buffer (100 mM Tris, pH 6.8, 4% SDS, in Laemmli-sodium dodecyl sulfate, 600 mM B-mercaptoethanol, 200 mM Dithiothreitol (DTT), and 20% glycerol). Equal amounts of protein per sample (5 or 10 ug) was loaded into a 4–20% gradient precast mini-PROTEAN TGX gel (Bio-Rad) and proteins were separated with 200V in 1x running buffer (24.76 mM Tris base, 190 mM glycine, 0.1% SDS). Proteins were transferred to a nitrocellulose membrane using the Trans-blot turbo system (Bio-Rad), mixed molecular weight protocol (2.5A, 25V for 7 min), followed by 1 hr blocking with LI-COR blocking buffer at a 1:1 ratio with TBS. Primary antibodies, including concentration and incubation times, are given in the Key Resources Table. All secondary antibodies were LI-COR, and incubated at 1:10,000 for 1 hr at room temperature. Imaging was performed on a LI-COR Odyssey machine on both the 680 and 800 channels.

## Serum-free primary astrocyte culture

Astrocytes were isolated from postnatal day 3–6 pups as described above and previously (*Kahanovitch et al., 2018*). Following elution, astrocyte cell number was determined, and 0.75–1.0 × 10$^5$ cells were plated on 13 mm glass coverslips in a 24-well plate. The coverslips were poly-l-ornithine treated and laminin-coated. Astrocytes were maintained in serum-free, defined media (50% Neurobasal media, 50% MEM, 1 mM sodium pyruvate, 2 mM glutamine, and 1x B27). On the first day post-plating, fresh media was added. On the third day post-plating, a complete media change was performed. Subsequent media changes occurred every 3–4 days. Astrocytes were collected at 7 and 14 days in vitro (DIV).

## Primary astrocyte culture experiments

Primary astrocyte cultures were utilized at 14 DIV for experiments. Exogenous BDNF (Promega) was applied in warmed media to a final concentration of 0 ng, 10 ng, 30 ng, or 100 ng for 24 hr. For scavenger experiments, a final concentration of 2 ug of TrkB-Fc (R and D Systems) was added to the wells in warmed media 60 min post-BDNF exposure. Equal volumes of warmed media as the TrkB-Fc condition was additionally added to controls.

## Primary astrocyte culture immunofluorescence

Astrocytes were fixed at 15 DIV, after experiments described above. First, pre-warmed paraformaldehyde (PFA) was added to the culture dish to a final concentration of 2% PFA, and incubated for 5 min at 37°C. This initial step was utilized to preserve any fine peripheral astrocyte processes that might be sensitive to cold temperatures. After incubation, cells were washed with cold PBS, followed by fixation with 4% PFA for 15 min at room temperature. Subsequently, cells were incubated for 1 hr in blocking buffer (10% goat serum, 0.3% Triton-X in PBS). Astrocyte filaments were visualized with GFAP, total membrane with Glast and Ezrin. Following primary antibody incubations, AlexaFlour 488, 546, and 647 were utilized to visualize the primary antibodies with 1 hr incubations. Prior to image acquisition, the experimenter was blinded to experimental conditions. Fluorescent images were acquired with an Olympus VS-120 system or Nikon A1 confocal.

## Primary astrocyte culture morphology analysis

The complexity of astrocytes following experiments described above was determined by utilizing the Shape Index, given as perimeter$^2$/area - 4$\pi$ (*Holt and Olsen, 2016*; *Matsutani and Yamamoto, 1997*). A perfect circle results in an index of 1, and increasingly complex cells have correspondingly larger indexes. Area and perimeter of the cells were determined manually using ImageJ 1.52b version software. Prior to quantification, experimenter was blinded to experimental conditions.

## In vivo astrocyte morphological analysis

Astrocytes were fluorescently labeled via AAV. Lck-GFP virus—pAAV.GfaABC1D.PI.Lck-GFP.SV40—was a gift from Baljit Khakh (Addgene viral prep # 105598-AAV5). AAV2/5 GfaABC1D.eGFP-Cre was obtained from UNC Gene Therapy Center Vector Core. AAV p2.1GfaABC1D.tdTomato was a gift from Baljit Khakh (Addgene viral prep # 44332-AAV5). Postnatal day 0–1 pups were intraventricularly injected with 2–3 µL 2.3 × 10$^8$ virus following hypothermia-induced anesthesia. For TrkB.T1 floxed experiments,. T1 fl/fl and WT pups were co-injected with eGFP-Cre and tdTomato virus. The injection site was determined following (*Chakrabarty et al., 2013*; *Kim et al., 2014*; *Shen et al., 2001*), with equidistance between the bregma and lambda sutures, 1 mm lateral from the midline, and 3 mm depth. Hamilton 10 µL syringes and 32G needles were used. Animals were collected at PND14, PND28-30 (referred to PND28 in manuscript), and PND 50–55 (referred to as PND 50 in manuscript). At time of collection, animals were deeply anesthetized with peritoneal injections of 100 mg/kg ketamine and intracardially perfused with PBS, followed by 4% PFA for 20 min. Brains were post-fixed for 72 hr, and subsequently sliced on Pelco Easislicer microtome at 100 µM sections. Experimenter was blinded to animal genotypes prior to image acquisition and analysis. Layer II/III motor cortex astrocytes were imaged on a Nikon A1 confocal with 40x oil immersion lens (OFN25) and 3x digital zoom. Z-stacks were acquired with 0.225 µM step sizes. Laser power and gain were adjusted for each individual astrocyte. Z-stacks were 3D reconstructed on Imaris x64 9.0.2, and surface reconstruction utilized to estimate astrocyte volume. Prior to quantification, experimenter was blinded to experimental conditions.

## Astrocyte and neuron counting

### Tissue Sectioning and Immunostaining

Wildtype and TrkB.T1 KO mice at PND 28 were collected via transcardial perfusions as described above, post-fixed for 72 hr, and sectioned into 50 µM slices. Sections containing the motor cortex were selected as detailed in *Franklin and Paxinos (2013)*. Sections were blocked for 1 hr in blocking buffer, followed by overnight incubation with primary anitbodies in diluted blocking buffer (1:3 of blocking buffer in PBS). Sections were then washed for 15 min (3x) in diluted blocking buffer before being incubated with secondary antibodies in diluted blocking buffer for 1.5 hr at room temperature

in the dark. Three additional 15 min washes in diluted blocking buffer followed the incubation in secondary antibodies. Finally, sections were incubated in DAPI (1:10,000) for five minutes before a last wash in PBS. Sections were mounted with Fluoromount medium (Cat#: F4680, Sigma) on glass slides (Cat #: 12-550-15, Fisher Scientific) and covered with cover glass (Cat#: 12-548-5E, Fisher Scientific). The edges of the slides were sealed with clear nail polish. Primary antibodies used: mouse NeuN (Cat#: MAB377, Millipore, 1:500) and rabbit Sox9 (Cat#: ab185966, Abcam, 1:500). Secondary antibodies used: mouse Texas Red (Cat#: T-6390, Invitrogen, 1:500) and AlexaFluor 488 (Cat#: A11008, Invitrogen, 1:500).

## Stereological Quantification

Three serial sagittal sections from each subject were collected via systematic random sampling (every tenth, 50 µm thick section/or three 50 µm sections spaced 500 µm apart). For each section, a 500 µm wide contour of the cortex above area CA2 of the hippocampus was drawn using a 4x objective lens and StereoInvestigator (MicroBrightField, Willinston, VT, USA.). The number of astrocytes and neurons in the cortex was assessed with the Optical Fractionator probe from Stereo Investigator (MicroBrightField, Willinston, VT, USA) and an Olympus BX51TRF motorized microscope (Olympus America). After delineation of the ROIs, StereoInvestigator systematically overlaid optical dissector grids that represented the individual fields for sampling/counting (grid size: 150 µm x 150 µm, counting frame size: 60 µm x 60 µm). Approximately 100 randomized sites were assessed to identify DAPI, Sox9+ and NeuN+ cells per animal. The number of cells per contour, average section thickness, section interval, and the number of sections were used to calculate the total number of cells within the sampled areas of the somatosensory cortex.

## Cortical thickness quantification

Serial sagittal sections used for stereological quantification were imaged at 5x magnification on a Zeiss Axio Observer D1 Inverted Phase Contrast Florescent Microscope using an Axiocam 506 Mono camera and Zen Blue Edition software (Carl Zeiss Microscopy GmbH, Göttingen, Germany). Cortical regions above hippocampal CA2 were imaged using Zeiss DAPI Filter Set 49 for cortical thickness determination. The stratum oriens of CA2 was used as a central marker for drawing segmented lines from layer I through layer VI in Fiji software (version 1.52 n), where segments would span from the most dorsal portion of the somatosensory region to the fiber tracts of the corpus callosum. Fiji ROI Manager measure tool assessed the length of each segment for determination of total length and segment length corresponding to each cortical layer.

## Primary neuron-astrocyte co-culture

Neurons were cultured from p0-1 mouse pups according to *Beaudoin et al. (2012)* and as described above with modifications. In brief, following cortical dissociation, microglia and oligodendrocytes were first removed with 10 µL incubation with $Cd11b^+$ and $Mbp^+$ microbeads for 10 min. The flow through was collected and used to further isolate neuronal populations utilizing Miltenyi Biotec's Neuron Isolation Kit. The cell suspension was incubated with 10 µL biotinylated antibodies for 10 min at 4°C, followed by a 10 min incubation with 15 µL anti-biotin microbeads. Cells were applied to a prepped LS column. Neurons were collected in the flow through of two washes. Neuronal cell number was determined, and $0.75–1.0 \times 10^5$ cells were plated on 13 mm glass coverslips in a 24-well plate. The coverslips were poly-l-lysine treated and laminin-coated. Neurons were maintained in neuronal maintenance media (*Beaudoin et al., 2012*) (Neurobasal media, 2 mM l-glutamine, and 1x B27). On the first day post-plating, 2 µM of araC was added to reduce non-neuronal contamination. On the second day post-plating, a media change was performed to remove araC. Subsequent media changes occurred every 3–4 days. At 3DIV $0.75–1.0 \times 10^5$ WT or TrkB.T1 KO astrocytes from p5 pups were plated on top of neurons.

## Co-cultured astrocyte morphology analysis

Cells were fixed at 12-14DIV, after co-culture describe above. Fixation was performed as described above. Subsequently, cells were incubated with 1 hr in blocking buffer (10% goat serum, 0.3% Triton-X in PBS). Glast and Gfap immunohistochemistry was performed as described above. Following primary antibody incubations, AlexaFlour 488 and 647 were used to visualize the primary antibodies

with 1 hr incubations at 1:500. Prior to image acquisition, the experimenter was blinded to experimental conditions. Confocal images were acquired on Nikon A1 confocal with 40x objective and two digital zoom. Astrocyte complexity was assessed by Sholl analysis plugin in FIJI software according to *Stogsdill et al. (2017)*. Briefly, for each individual astrocyte, a line was drawn from the soma out to the furthest cellular process. Sholl radii were set at starting 10 μm from the start line with 1 μm increasing increments. Subsequently, the data were binned at 10 μm intervals in Excel.

## Neuron synapse quantification

Cells were collected at 8–9 DIV and 12–14 for synapse quantification. Cells were fixed as described above. Subsequently, cells were incubated for 1 hr in blocking buffer (10% goat serum, 0.3% Triton-X in PBS). Excitatory synapses were visualized with presynaptic marker VGLUT1 and with postsynaptic marker PSD95. In a subset of experiments, synapses were visualized with presynaptic marker Bassoon and with postsynaptic marker Homer1. Following primary antibody incubations, AlexaFlour 488 and 647 were utilized to visualize the primary antibodies with 1 hr incubations at 1:500. These secondaries were chosen for their excitation/emission spectrum, which demonstrate no overlap. Therefore, co-localization analysis of pre- and post-synaptic markers can be utilized with confidence of true co-localization. Prior to image acquisition, the experimenter was blinded to experimental conditions. Confocal images were acquired on Nikon A1 confocal with 40x objective and three digital zoom. Care was taken to ensure each individual neuron imaged was equidistant from other neurons and astrocytes. Co-localization, and therefore synapse number, was determined utilizing Puncta Analysis FIJI plug-in (*Ippolito and Eroglu, 2010*; *Stogsdill et al., 2017*). Briefly, three random 10 μm sections of the neuronal dendrite that were at least 10 μm away from neuronal soma were analyzed using Puncta Analysis software. The three regions of interest were averaged per cell in Excel prior to performing statistics. At least three neurons per coverslip, and two coverslips per culture were examined.

## Neuronal whole-cell electrophysiology and mini analysis

Neuronal astrocyte co-cultures (neurons 10–14 DIV) were used for evaluation neuronal spontaneous miniature excitatory post-synaptic currents. Experimenter was blinded to condition prior to recording. Briefly, coverslips were transferred to a recording chamber on an inverted Zeiss Observer.D1 microscope and continuously perfused (2 mL/minute at room temperature) with aCSF containing (in mM), NaCl 116, KCl 4.5, $MgCl_2$ 0.8, $NaHCO_3$ 26.2, glucose 11.1, HEPES 5.0). Patch pipettes were made from thin-walled (outer diameter 1.5 mm, inner diameter 1.12 mm) borosilicate glass (TW150F-4, WPI, FL) and had resistances of 4–8 MΩ when filled with Cs-gluconate solution pipette solution contained (in mM) 120 Cs-gluconate, 17.5 CsCl, 10 Na-Hepes, 4 Mg-ATP, 0.4 Na_GTP, 10 Na2-creatine phosphate, 0.2 Na-EGTA; 290–300 mOsm; pH 7.3 (*Calfa et al., 2015*). Current recordings were obtained from neurons with an Axopatch 200A amplifier (Axon Instruments) signals were low-pass filtered at 1 kHz and were digitized on-line at 10–20 kHz using a Digidata 1320 digitizing board (Axon Instruments). Data acquisition and storage were conducted with the use of pClamp 10.2 (Axon Instruments). Cell capacitances and series resistances were measured directly from the amplifier, series resistance compensation adjusted to 80% to reduce voltage errors. Cells with a series resistance >12 were omitted from study to reduce voltage errors. Spontaneous miniature EPSC's in 1 μM TTX (Tocris) were recorded at −60 mV. Prior to analysis, experimenter was again blinded to condition. Mini event amplitude and frequency were analyzed using MiniAnalysis (Synaptosoft).

## Statistical analysis

To determine statistical significance, Origin and Graphpad Prism were utilized. All data is represented as mean + /- SEM, with n's indicated where appropriate. D'Agostino-Pearson normality test was performed to determine the normality distribution of each data set, and outliers were determined via GraphPad Prism's ROUT method. Student's t-tests were performed, or Mann Whitney U tests for nonparametric data, for all data in which only one comparison was needed. One-way ANOVAs, or Kruskal-Wallis test for nonparametric data, followed by Tukey's post-hoc test performed for all data with multiple comparisons.

## Acknowledgements

The authors wish to thank Lara Ianov and CIRC Neurodevelopmental Bioinformatics Initiative for their help in processing the RNA-sequencing and Dr. Lina Tessorallo and colleagues for the kind gift of the TrkB.T1 KO and TrkB.T1 floxed mouse model. This work was funded by the National Institutes of Health NINDS R01NS075062 (MO) and F31NS100259 (LMH).

## Additional information

### Funding

| Funder | Grant reference number | Author |
| --- | --- | --- |
| National Institute of Neurological Disorders and Stroke | F31NS100259 | Leanne M Holt |
| National Institute of Neurological Disorders and Stroke | R01NS075062 | Michelle Olsen |

The funders had no role in study design, data collection and interpretation, or the decision to submit the work for publication.

### Author contributions

Leanne M Holt, Conceptualization, Data curation, Formal analysis, Funding acquisition, Validation, Investigation, Visualization, Methodology, Writing—original draft, Project administration, Writing—review and editing; Raymundo D Hernandez, Beatriz Torres Ceja, Data curation, Formal analysis, Investigation; Natasha L Pacheco, Data curation, Formal analysis; Muhannah Hossain, Investigation; Michelle L Olsen, Conceptualization, Resources, Data curation, Formal analysis, Supervision, Funding acquisition, Validation, Investigation, Visualization, Methodology, Writing—original draft, Project administration, Writing—review and editing

### Author ORCIDs

Leanne M Holt ⓘD https://orcid.org/0000-0003-0954-2298
Natasha L Pacheco ⓘD http://orcid.org/0000-0002-9617-8887
Beatriz Torres Ceja ⓘD https://orcid.org/0000-0001-5235-4677
Michelle L Olsen ⓘD https://orcid.org/0000-0003-1394-664X

### Ethics

Animal experimentation: All experiments were performed according to NIH guidelines and with approval from the Animal Care and Use Committee of the University of Alabama at Birmingham (#20650) and Virginia Polytechnic Institute and State University (#17-012). Every effort was made to minimize pain and discomfort.

### Decision letter and Author response

Decision letter https://doi.org/10.7554/eLife.44667.sa1
Author response https://doi.org/10.7554/eLife.44667.sa2

## Additional files

### Supplementary files

• Transparent reporting form

### Data availability

Sequencing data have been deposited in GEO under accession code GSE122176.

The following dataset was generated:

| | | | Database and Identifier |
| --- | --- | --- | --- |

| Author(s) | Year | Dataset title | Dataset URL | |
|---|---|---|---|---|
| Pacheco NL, Olsen ML | 2018 | Data from: Astrocyte morphogenesis is dependent on BDNF signaling via astrocytic TrkB.T1 | https://www.ncbi.nlm.nih.gov/geo/query/acc.cgi?acc=GSE122176 | NCBI Gene Expression Omnibus, GSE122176 |

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
