## [Decision Letter]

Thank you for submitting your article "BDNF/TrkB.T1 signaling is a novel mechanism for astrocyte morphological maturation" for consideration by *eLife*. Your article has been reviewed by three peer reviewers, and the evaluation has been overseen by a guest Reviewing Editor and Didier Stainier as the Senior Editor. The following individual involved in the review of your submission has agreed to reveal their identity: Marc Freeman (Reviewer #1).

The reviewers have discussed the reviews with one another and the Reviewing Editor has drafted this decision to help you prepare a revised submission. Please aim to submit the revised version within two months. If you feel this is an unrealistic amount of work for 2 months, please send back within a week an action plan that outlines what you consider reasonable revisions and the editors and reviewers will discuss it.

Summary:

In this study, the authors report a role for BDNF/TrkB.T1 signaling in astrocyte morphogenesis and morphological complexity. The authors show data that loss of TrkB.T1 impacts astrocyte maturation and dysregulates expression of genes involved in astrocyte function. Additionally, the authors show astrocytes lacking TrkB.T1 do not support normal synaptogenesis. Key data supporting their model includes astrocyte-specific expression of TrkB.T1 in vitro and in vivo, a developmental upregulation of TrkB.T1, KO data in vitro and in vivo supporting a role for TrkB.T1 in promoting astrocyte morphological complexity (as well as Fc-depletion of BDNF), changes in the expression of key astrocyte functional genes, and a reduced capacity of these astrocytes to promote post-synaptic development. The three reviewers and the reviewing editor all found the data convincing and experiments well conducted. However, they also agreed that to fully support the claims of this study, additional electrophysiological and in vivo data are required. Moreover, the reviewers pointed out several necessary clarifications and textual edits to ensure that the data presented and the conclusions drawn from the are in line.

Essential revisions:

Major concerns to be addressed during revision:

1) Electrophysiological analyses are needed to confirm the claim that loss of TrkB in astrocytes alters the ability of astrocytes to induce synapse formation and enhance synaptic function in vitro.

2) The decrease in astrocyte volume/complexity in TrkB.T KO mice in vivo requires more investigation. Is this difference maintained in later ages? Does it lead to reduced neuropil infiltration by astrocytes and/or generation of more astrocytes due to reduced territory sizes? For example, is there still a deficit in astrocyte volume at older ages, such as P60, when astrocyte volume has stabilized (see Morel et al., 2014)? If the volume of all astrocytes in the cortex is decreased, this would predict that there is now an increase in the volume of the extracellular space. If so, this would be expected to have a significant impact on neuronal activity, for example by altering potassium uptake and neurotransmitter recycling. Are the mice reported to show any behaviors that would correlate with this, for example seizures? Alternatively, are there more astrocytes in the KO cortex that fill in the space? This could be quantified by counting the number of *Sox9* positive nuclei in the KO and WT (see Sun et al., Journal Neuroscience, 2017).

3) A key limitation of the study is the utilization of a global TrkB.T1 knockout mouse. In order to draw conclusions on the role of this protein in astrocytes in vivo, the authors should manipulate TrkBT1 expression only in astrocytes and test the effects on astrocyte morphogenesis. Targeting of astrocytes for gene manipulation by viral or electroporation approaches provide the means for doing this without the need for generating conditional KOs. So, we believe this experiment can be conducted within a reasonable time frame during revisions.

4) The data provided is not strong enough to firmly link impaired synaptogenic signaling by sparcl1 and sparc signaling to the phenotypes associated with loss of TrkB in astrocytes. We ask the authors to remove these experiments or conduct new experiments that will causally link sparcl1/sparc-signaling as a potential mechanism. However, the authors should also consider that astrocytes secrete a number of synapse modulating proteins and those may also be at play here. They should be able to rule them in or out before landing on sparcl1 and sparc. But we also agreed that this may be beyond the scope of one study. Therefore, it is recommended that you focus on the top three points in the revisions and leave this aspect for future research.

Textual Clarifications/Changes Required:

1) It is difficult to understand exactly how astrocyte complexity was quantified. Please provide more clarity on this point. It is hard to tell from the images how dramatic the change is with BDNF manipulation. This is difficult to do, but is a key part of the study.

2) Claims to 'first time' and 'novelty' should be avoided. For example, the last paragraph of the Introduction states this is the first time the TrkB receptor has been detected in astrocytes, however there are many reports of this in the literature e.g. Rose et al., 2003, Ohira et al., 2005, Ohira et al., 2007, Matyas et al., 2017. Further, in two of these papers a role for truncated TrkB in regulating astrocyte morphology in vitro and in brain slices (fine process motility) was investigated (Ohira et al., 2005 and 2007). While we appreciate the current study uses a more physiological astrocyte model (serum free culture, postnatal astrocyte isolation), and is looking at overall astrocyte growth during postnatal maturation, the 'first time' and 'novelty' claims cannot be supported due to these earlier studies.

3) The utilization of a global TrkB.T1 knockout mouse does not allow support for the claim "the majority effects within the global TrkB.T1 KO mouse may be attributable to astrocytes." In cortical circuit development, neuron generation happens before astrocyte generation, which is a quite different from the cell-specific astrocyte culture system employed in this study. Using cell-specific astrocyte cultures, the authors assessed the impact of TrkB.T1 on astrocytes, however it does not necessarily represent the systemic effects of TrkB.T1 on astrocytes. Please carefully go through the manuscript to ensure that no overarching comments that cannot be fully supported by the experiments conducted are present.

4) Related to the point raised above, the authors show impairments of morphological maturation of astrocytes in the TrkB.T1 KO mouse compared to that in WT mouse in vivo; however, the impact of morphological complexity demonstrated in TrkB.T1 KO astrocyte cultures only occurred under external BDNF application. This is an important point that is not addressed and may suggest various processes are in play. Please discuss/clarify.

5) In the WT astrocyte cultures (Figure 2), the enhancement of TrkB.T1 mRNA expression was detected at DIV14; however, the astrocyte morphological complexity in TrkB WT and TrkB.T1 KO astrocyte cultures had no difference under the control treatment at the same DIV14 (Figure 3). The authors should comment on this point. Moreover, in separate experiments the authors assessed synaptic development 5 days after seeding WT or TrkB.T1 KO astrocytes on the top of neurons (Figure 5). Given the various timeframes examined and the varying results, it is difficult to conclude that the impact of excitatory synaptic development measured at DIV8-9 in Figure 5 resulted from astrocytes lacking TrkB.T1 in their co-culture experiment. Explain the differences between the timelines utilized between different experiments.

6) It is mentioned in the Discussion that both excitatory and inhibitory synapses were affected, with TrkB manipulation in astrocytes, but only data for excitatory synapses is shown. Please correct this error.

7) The authors collected samples from neurons, astrocytes or microglia at various time points, but the indication of the expression of RNA or protein at these time points were unclear in the figures and figure legends. Please label the time points on the figure or make a developmental timeline and pinpoint the time window for each experiment in each figure.

8) There should be some discussion as to why BDNF only has an effect on astrocyte morphology at 30ng/ml, and not 100ng/ml. Is BDNF acting through the truncated TrkB receptor known to have a bell-shaped dose response?

---

## [Author Response]

Essential revisions:Major concerns to be addressed during revision:1) Electrophysiological analyses are needed to confirm the claim that loss of TrkB in astrocytes alters the ability of astrocytes to induce synapse formation and enhance synaptic function in vitro.

Thank you for this suggestion. We agree that electrophysiological analysis is necessary to support our in vitro synaptic data. New electrophysiological experiments were performed and are now added to Figure 6. These data support the notion that TrkB.T1 astrocytes do not support normal excitatory synaptogenesis.

*2) The decrease in astrocyte volume/complexity in TrkB.T KO mice* in vivo *requires more investigation. Is this difference maintained in later ages? Does it lead to reduced neuropil infiltration by astrocytes and/or generation of more astrocytes due to reduced territory sizes? For example, is there still a deficit in astrocyte volume at older ages, such as P60, when astrocyte volume has stabilized (see Morel et al., 2014)? If the volume of all astrocytes in the cortex is decreased, this would predict that there is now an increase in the volume of the extracellular space. If so, this would be expected to have a significant impact on neuronal activity, for example by altering potassium uptake and neurotransmitter recycling. Are the mice reported to show any behaviors that would correlate with this, for example seizures? Alternatively, are there more astrocytes in the KO cortex that fill in the space? This could be quantified by counting the number of Sox9 positive nuclei in the KO and WT (see Sun et al., Journal Neuroscience, 2017).*

Thank you. We have broken the reviewer comments into separate sections in order to clearly address each point.

A) Change in astrocyte morphological complexity at later ages.

We agree that this is an important question. We have now included astrocyte morphological complexity analysis at PND 50 (Morel, 2014) in Figure 4. These data demonstrate a significant 20% reduction in astrocyte morphological complexity at P50.

B) Change in extracellular space / astrocyte number as compensation.

The reviewers bring up a very interesting point. As suggested, we performed stereological microscopy to determine the number of astrocytes and neurons in the TrkB.T1 KO mice. We have included new experimental analysis of astrocyte cell numbers as suggested by the reviewer and included the following in the Discussion:

“We did not observe a difference in astrocyte number by stereological microscopy across the motor cortex, nor did we observe a difference cortical motor thickness. […] Studies in astrocyte specific TrkB.T1 KO are needed to determine if loss of TrkB.T1 in astrocytes contribute to this aberrant behavior.”

Additionally, we have found no publications investigating seizure susceptibility in the TrkB.T1 KO mice, and little investigation of neuronal activity in this model. However, published reports (Carim-Todd, 2009) demonstrate that the mice do exhibit behaviors associated with increased anxiety. We feel that the susceptibility of the mice to seizures is outside of the scope of this manuscript, however behavioral assessment in the conditional-TrkB.T1 KO mouse we have generated are an area we plan to investigate.

3) A key limitation of the study is the utilization of a global TrkB.T1 knockout mouse. In order to draw conclusions on the role of this protein in astrocytes in vivo, the authors should manipulate TrkBT1 expression only in astrocytes and test the effects on astrocyte morphogenesis. Targeting of astrocytes for gene manipulation by viral or electroporation approaches provide the means for doing this without the need for generating conditional KOs. So, we believe this experiment can be conducted within a reasonable time frame during revisions.

Thank you. We agree with the reviewers that the utilization of the global TrkB.T1 KO model is a key limitation to the study, and that astrocyte-specific loss of TrkB.T1 in an otherwise normal environment is an important experiment. We therefore performed three experiments in parallel, presented in a new Figure 5, to address this concern. These data unambiguously indicate that the loss of TrkB.T1 in astrocytes alone is sufficient to result in reduced astrocyte morphological complexity. We have outlined the experiments and findings below:

A) We assessed the complexity of astrocyte morphology from WT and TrkB.T1 KO astrocytes co-cultured with WT neurons. Using both Glast to demarcate the membrane (and therefore total astrocyte complexity) and Gfap to demarcate the processes, we found that TrkB.T1 KO astrocytes demonstrated decreased morphological complexity in comparison to WT astrocytes.

B) We utilized viral-mediated Cre-recombinase to knockdown TrkB.T1 specifically in astrocytes. Using a previously generated TrkB.T1 floxed mouse line and our ICV injection paradigm, WT and. T1^fl/fl^ mice were co-injected with an astrocyte-specific AAV driving Cre recombinase with an EGFP florescent reporter and a control astrocyte-specific AAV driving only a tdTomato fluorescent reporter. Astrocyte morphology was assessed in both egfp^+^(Cre) and tdtom^+^(control) astrocytes in both WT and.T1fl/fl mice. Egfp+(Cre+) astrocytes from. T1fl/fl animals exhibited decreased volume compared to Egfp+(cre+) WT astrocytes. Importantly, we found no difference between tdtom+(control).T1fl/fl and WT astrocytes.

C) We generated an astrocyte-specific TrkB.T1 knockout model. Using the same TrkB.T1 floxed line as above, we crossed these mice to astrocyte targeted Aldh1l1-CreERT2 mice. Following tamoxifen injections between P8-11, we found loss of the TrkB.T1 receptor specifically in astrocytes. Animals were injected with the AAV2/5 Gfap-egfp virus, TAM injected, and astrocyte morphology assessed. We again found decreased astrocyte complexity in the cKO animals compared to cWT littermates.

4) The data provided is not strong enough to firmly link impaired synaptogenic signaling by sparcl1 and sparc signaling to the phenotypes associated with loss of TrkB in astrocytes. We ask the authors to remove these experiments or conduct new experiments that will causally link sparcl1/sparc-signaling as a potential mechanism. However, the authors should also consider that astrocytes secrete a number of synapse modulating proteins and those may also be at play here. They should be able to rule them in or out before landing on sparcl1 and sparc. But we also agreed that this may be beyond the scope of one study. Therefore, it is recommended that you focus on the top three points in the revisions and leave this aspect for future research.

Thank you. As the reviewers suggested, we have removed the sparcl/sparc data from the manuscript.

Textual Clarifications/Changes Required:1) It is difficult to understand exactly how astrocyte complexity was quantified. Please provide more clarity on this point. It is hard to tell from the images how dramatic the change is with BDNF manipulation. This is difficult to do, but is a key part of the study.

Thank you for this highlighting this ambiguity in the manuscript. We have added the following sentence to the Results section clarify how the Shape Index quantifies cellular complexity.

“We utilized the Shape Index (SI) to examine in vitro astrocyte complexity; this equation, given as perimeter^2^/area – 4π, relates area and perimeter of a cell to a circle such that a perfect circle receives an SI of 0, and increasingly complex cells have correspondingly larger SI values.”

Analysis of shape index data indicate a > 2-fold increase in morphological complexity. Cumulative frequency also shows the rightward shift in morphological complexity in astrocytes treated with BDNF. The images of astrocytes presented are representative. The 30 ng BDNF image demonstrates a striking difference in astrocyte morphology compared with the 4 other images in the figure. We now provide additional images in Figure 3—figure supplement 1 with example tracing of individual astrocytes used for data analysis in Image J.

*2) Claims to 'first time' and 'novelty' should be avoided. For example, the last paragraph of the Introduction states this is the first time the TrkB receptor has been detected in astrocytes, however there are many reports of this in the literature e.g. Rose et al., 2003, Ohira et al., 2005, Ohira et al., 2007, Matyas et al., 2017. Further, in two of these papers a role for truncated TrkB in regulating astrocyte morphology* in vitro *and in brain slices (fine process motility) was investigated (Ohira et al., 2005 and 2007). While we appreciate the current study uses a more physiological astrocyte model (serum free culture, postnatal astrocyte isolation), and is looking at overall astrocyte growth during postnatal maturation, the 'first time' and 'novelty' claims cannot be supported due to these earlier studies.*

Thank you. We have carefully reviewed the manuscript and have striven to remove language that suggests incorrect novelty. However, we do feel that our data does represent novel findings in regards to TrkB receptor expression in astrocyte development. In particular, we are indeed the first examine TrkB receptor expression specifically in astrocytes across cortical development. To ensure we do not overgeneralize, we have included the following in the Discussion:

“BDNF’s role in CNS growth and maturation has been intensely studied, with a particular emphasis on the full length receptor and neuronal function. […] Publicly available RNA sequencing databases corroborate our data (Zhang et al., 2014), and, in fact, additionally demonstrate that human astrocytes express the highest levels of Ntrk2 (Kelley et al., 2018; Zhang et al., 2016).”

3) The utilization of a global TrkB.T1 knockout mouse does not allow support for the claim "the majority effects within the global TrkB.T1 KO mouse may be attributable to astrocytes." In cortical circuit development, neuron generation happens before astrocyte generation, which is a quite different from the cell-specific astrocyte culture system employed in this study. Using cell-specific astrocyte cultures, the authors assessed the impact of TrkB.T1 on astrocytes, however it does not necessarily represent the systemic effects of TrkB.T1 on astrocytes. Please carefully go through the manuscript to ensure that no overarching comments that cannot be fully supported by the experiments conducted are present.

Thank you. We have removed this section. We have reviewed the manuscript and have striven to remove overgeneralization of our results.

4) Related to the point raised above, the authors show impairments of morphological maturation of astrocytes in the TrkB.T1 KO mouse compared to that in WT mouse in vivo; however, the impact of morphological complexity demonstrated in TrkB.T1 KO astrocyte cultures only occurred under external BDNF application. This is an important point that is not addressed and may suggest various processes are in play. Please discuss/clarify.

Thank you. We agree that this is an important and complicated point. We have addressed this within the Discussion of the manuscript. We feel that our data can be explained, in part, due to the differences in in vitro and in vivo astrocytes. It is well characterized that astrocyte cultures are not fully representative of in vivo astrocytes. While many of these differences can be attributed to the presence of serum in traditional astrocyte cultures (Foo et al., 2011), culturing astrocytes in isolation also removes them from the influence of other cellular populations. The loss of various signaling molecules from other CNS populations, and in this context, the lack of BDNF present in the media, may explain why the in vitro WT and TrkB.T1 KO astrocyte morphology does not differ except under external application.

5) In the WT astrocyte cultures (Figure 2), the enhancement of TrkB.T1 mRNA expression was detected at DIV14; however, the astrocyte morphological complexity in TrkB WT and TrkB.T1 KO astrocyte cultures had no difference under the control treatment at the same DIV14 (Figure 3). The authors should comment on this point.

The reviewers bring up an interesting point. As mentioned in response to #4, culturing astrocytes removes them from communication with other cellular populations, and thus under control conditions the media will not contain BDNF. Under this assumption, with no stimulation from BDNF, it would be predicted that control conditions between WT and TrkB.T1 KO astrocytes would not exhibit differences in morphological complexity. Only in the presence of the TrkB.T1 receptor’s ligand, BDNF, would WT astrocytes be signaled to increase in morphological complexity. Our assertion that TrkB.T1 increases astrocyte morphological complexity only with BDNF stimulation is also supported by our TrkB-Fc scavenge data, whereby removing the exogenously applied BDNF from the media results in astrocytes with morphological complexity similar to control astrocytes.

Moreover, in separate experiments the authors assessed synaptic development 5 days after seeding WT or TrkB.T1 KO astrocytes on the top of neurons (Figure 5). Given the various timeframes examined and the varying results, it is difficult to conclude that the impact of excitatory synaptic development measured at DIV8-9 in Figure 5 resulted from astrocytes lacking TrkB.T1 in their co-culture experiment. Explain the differences between the timelines utilized between different experiments.

While our data does demonstrate an upregulation of TrkB.T1 in the astrocytes at 14DIV, we feel that it is not indicative of a lack of TrkB.T1 receptor expression in 7DIV astrocytes. We highlighted this observation in the manuscript only as it is on a similar to the upregulation we found between PND 7 and PND 28 astrocytes, and thus suggests an intrinsic regulation of *Ntrk2* expression in astrocytes. In fact, preliminary data from our lab indicates that *Ntrk2* RNA expression is in the top 50 most highly expressed genes found in acutely isolated wildtype astrocytes from postnatal day 6-60. Thus, BDNF may exert effects on astrocytes across their development and maturation. We feel that the impact on excitatory synaptic development measured at DIV8-9 can only be explained by the one manipulated variable, the loss of the truncated receptor in astrocytes. However, to address this concern, we performed synaptic quantification in wildtype neurons co-cultured with astrocytes for a longer, more extended timeframe with collection at 12-14DIV. As demonstrated in Author response image 1, we found the same change in excitatory synapse number. And, in fact, we found no significant difference in synaptic numbers between the two experimental designs (black: 12-14 DIV and red 8-9 DIV). We have therefore collapsed these datasets into one figure and expanded the time of collection in the manuscript.

6) It is mentioned in the Discussion that both excitatory and inhibitory synapses were affected, with TrkB manipulation in astrocytes, but only data for excitatory synapses is shown. Please correct this error.

Thank you. This has been corrected in the manuscript.

7) The authors collected samples from neurons, astrocytes or microglia at various time points, but the indication of the expression of RNA or protein at these time points were unclear in the figures and figure legends. Please label the time points on the figure or make a developmental timeline and pinpoint the time window for each experiment in each figure.

Thank you for highlighting the ambiguity of the figure. We have updated both the figures and figure legends to make the time line of assessment more clear.

8) There should be some discussion as to why BDNF only has an effect on astrocyte morphology at 30ng/ml, and not 100ng/ml. Is BDNF acting through the truncated TrkB receptor known to have a bell-shaped dose response?

The reviewers bring up a very interesting point. To the authors’ knowledge, thus far experimental determination of a dose-response of TrkB.T1 has not been performed. The authors chose the experimental BDNF concentrations as they have been used previously to examine BDNF’s effects on cultured cellular populations. We have added the following into the Discussion section of the manuscript:

“A potential interaction and/or regulation of RhoGTPases by TrkB.T1 may also offer insight into the BDNF dose-specific response we observed in cultured astrocytes. […] However, our data suggests TrkB.T1 plays an active, direct role in astrocyte biology.”